# Ketohexokinase-A acts as a nuclear protein kinase that mediates fructose-induced metastasis in breast cancer

Jiyoung Kim [1,2,3,6], Jengmin Kang [1,2,3,6], Ye-Lim Kang[1,2], Jongmin Woo [2], Youngsoo Kim [2,4], June Huh [5] & Jong-Wan Park [1,2,3 ✉]

Harmful effects of high fructose intake on health have been widely reported. Although fructose is known to promote cancer, little is known about the underlying mechanisms. Here, we found that fructose triggers breast cancer metastasis through the ketohexokinase-A signaling pathway. Molecular experiments showed that ketohexokinase-A, rather than ketohexokinase-C, is necessary and sufficient for fructose-induced cell invasion. Ketohexokinase-A-overexpressing breast cancer was found to be highly metastatic in fructose-fed mice. Mechanistically, cytoplasmic ketohexokinase-A enters into the nucleus during fructose stimulation, which is mediated by LRRC59 and KPNB1. In the nucleus, ketohexokinase-A phosphorylates YWHAH at Ser25 and the YWHAH recruits SLUG to the *CDH1* promoter, which triggers cell migration. This study provides the effect of nutrition on breast cancer metastasis. High intake of fructose should be restricted in cancer patients to reduce the risk of metastasis. From a therapeutic perspective, the ketohexokinase-A signaling pathway could be a potential target to prevent cancer metastasis.

[1] Department of Pharmacology, Seoul National University College of Medicine, Daehak-ro, Jongno-gu, Seoul 03080, Korea. [2] Department of Biomedical Science, BK21-plus education program, Seoul National University College of Medicine, Daehak-ro, Jongno-gu, Seoul 03080, Korea. [3] Cancer Research Institute and Ischemic/Hypoxic Disease Institute, Seoul National University College of Medicine, Daehak-ro, Jongno-gu, Seoul 03080, Korea. [4] Department of Biomedical Engineering, Seoul National University College of Medicine, Daehak-ro, Jongno-gu, Seoul 03080, Korea. [5] Department of Chemical and Biological Engineering, Korea University, Anam-ro, Seongbuk-gu, Seoul 02841, Korea. [6] These authors contributed equally: Jiyoung Kim, Jengmin Kang. ✉email: parkjw@snu.ac.kr

Fructose naturally occurs in fruits and is a popular sweetener for cooking and baking[1]. Many reports have warned against excessive intake of fructose because it aggravates obesity, diabetes, and hepatic steatosis[2–4]. Aside from metabolic disorders, high consumption of fructose is associated with high risks of endometrial[5], breast[6], esophageal[7], small intestine[7], pleural[7], and colorectal cancers[8]. Fructose-induced carcinogenesis is considered to be attributed to altered energy metabolism, increased oxidative stress, and inflammation[9–11].

Glucose provides energy and intermediate metabolites required for amino acid and nucleic acid syntheses. The metabolism of cancer cells is reprogrammed to consume large quantities of glucose, resulting in glucose depletion in tumor microenvironments. To compensate for insufficient glucose, cancer cells use fructose as an alternative energy source. For instance, the fructose flux is increased in pancreatic cancer cells, which accelerates nucleic acid synthesis through the non-oxidative pentose phosphate pathway[12]. Besides cell proliferation, fructose is also associated with cancer metastasis. Fructose promotes colon cancer metastasis toward the liver via the ketohexokinase–aldolase B pathway[11]. In other cancers, fructose has been also reported to increase metastatic potential[13–16]. However, the precise mechanism underlying the metastogenic action of fructose remains unclear.

Ketohexokinase (KHK) is the enzyme that converts fructose to fructose-1-phosphate at the rate-limiting first step of fructose metabolism. The KHK gene expresses two isoforms, KHK-A and KHK-C, resulting from alternative splicing of exon 3. KHK-C, which is exclusively expressed in a few tissues, including the liver, drives the aforementioned metabolism. Conversely, KHK-A is not believed to contribute to this metabolism because its enzymatic function is very low at the physiological level of fructose. Nonetheless, as KHK-A is ubiquitously expressed in most tissues, KHK-A is believed to have some biological functions other than energy metabolism[17,18]. Recently, an interesting role of KHK-A was reported in hepatocellular carcinoma. KHK-A facilitates nucleic acid synthesis by directly phosphorylating phosphoribosyl pyrophosphate synthetase 1 (PRPS1), thereby promoting cell proliferation[19]. Considering its role as a protein kinase, KHK-A is expected to participate in diverse signaling pathways, which remain to be explored.

Breast-cancer cells highly express the fructose transporter GLUT5[20], and metastasize in response to high fructose[16]. However, little is known about the molecular mechanism underlying fructose-induced metastasis in breast cancer. Herein, we tested the possibility that KHK-A mediates breast-cancer metastasis in response to fructose. Upon fructose stimulation, LRRC59, in concert with KPNB1, transported KHK-A to the nucleus, where KHK-A phosphorylated YWHAH at Ser25. Phosphorylated YWHAH promoted SLUG to repress the CDH1 gene, thereby leading to cell invasion in breast cancer. This work provides the molecular mechanism underlying fructose-induced cancer metastasis.

## Results

**Fructose stimulates cell invasion in breast cancer**. We first examined which cancer cells became more invasive in response to fructose. Compared with saline or glucose, fructose facilitated Matrigel invasion in cancer cell lines derived from the breast, brain, lung, pancreas, colon, prostate, and uterine cervix, but not in those from the ovary, kidney, bone, and liver (Fig. 1a and Supplementary Fig. 1). To examine the role of KHK in this process, we measured the cellular levels of two KHK isoforms in various cancer cell lines. Most cell lines except HepG2 predominantly expressed KHK-A (Fig. 1b). Informatics analyses

using TCGA revealed that KHK-A, rather than KHK-C, is predominantly expressed in most human cancers, including breast cancer (Fig. 1c and Supplementary Fig. 2). We next explored the metastogenic role of KHK-A in breast cancer because cell invasion in response to fructose was strongly shown in all three breast-cancer cell lines. As glucose as well as fructose could affect cell behavior[21,22], we performed the invasion assay under the same concentration (finally 5 mM) of glucose or fructose. Fructose robustly stimulated cell invasion, but glucose at such a concentration showed a marginal effect on the invasion (Supplementary Fig. 3a). To examine which isoform was responsible for fructose-induced cell invasion, we silenced KHK-A or KHK-C using siRNAs that target isotype-specific sequences in exon 3. Consequently, fructose-induced invasion was attenuated by silencing KHK-A. Moreover, cell invasion was augmented by KHK-A overexpression (Fig. 1d and Supplementary Fig. 3b). To examine the structure of cytoskeletons, F-actin was stained with phalloidin. During fructose incubation, MDA-MB-231 cells were morphologically altered with enhanced front–rear polarization and F-actin rearrangement (Fig. 1e and Supplementary Fig. 3c), which indicates that cells underwent the epithelial–mesenchymal transition (EMT). Of the representative EMT markers, the cell-adhesion molecule CDH1 (alternatively named E cadherin) was substantially suppressed by fructose, and this effect was augmented by KHK-A overexpression (Fig. 1f and Supplementary Fig. 3d). Hence, our results suggest that fructose triggers EMT in breast cancer, which may be mediated by KHK-A. Given that KHK-A is known to be less active in fructose utilization, we needed to determine whether the fructose-induced invasion was attributable to the enzymatic function of KHK-A. Under fructose stimulation, both the cytoskeletal rearrangement and cell invasion in MDA-MB-231 cells were attenuated by a KHK-specific kinase inhibitor pyrimidinopyrimidine (Pypy) (Fig. 1g, h and Supplementary Fig. 3e). We also checked whether KHK-A participates in the fructose flux by measuring the fructose-1-phosphate and ROS. As expected, their levels were enhanced by KHK-C, but not by KHK-A (Supplementary Fig. 3f, g). These results suggest that KHK-A promotes cell invasion due to its kinase function regardless of fructose metabolism.

On the other hand, ALDOB and ALOX12 were previously reported to play roles in fructose-induced metastasis in colon[11] and breast cancers[15]. Therefore, we also investigated whether these proteins were involved in the fructose-stimulated invasion of breast-cancer cells. However, the knockdown of either ALDOB or ALOX12 failed to attenuate cell invasion upon fructose stimulation (Supplementary Fig. 4a, b). About inconsistent results with previous reports, we concluded that it is because of using different types of cell line or experiment under different circumstances cause alteration of cellular metabolism. As mentioned, KHK-A was reported to promote hepatocellular carcinogenesis by phosphorylating PRPS1[19]. Therefore, we explored the possibility that PRPS1 mediates KHK-A action on cell invasion. Even while PRPS1 was knocked down, MDA-MB-231 cells showed good invasiveness in response to fructose (Supplementary Fig. 4c). Taken together, these data indicate that KHK-A seems to promote cell invasion.

**KHK-A potentiates the fructose-induced metastasis**. To evaluate whether the KHK-A promotes breast-cancer metastasis in vivo, we implanted the murine MTV-TM-011 cell lines in the mammary fat pads of mice. The cell lines were established to stably express KHK-A or KHK-C/sh-Khk-a with luciferase and. Luciferase activity, KHK-A/C expressions, and knockdown of mouse KHK-A were verified as shown in Supplementary Figs. 5a and 6a. The experimental schedules are summarized in Fig. 2a

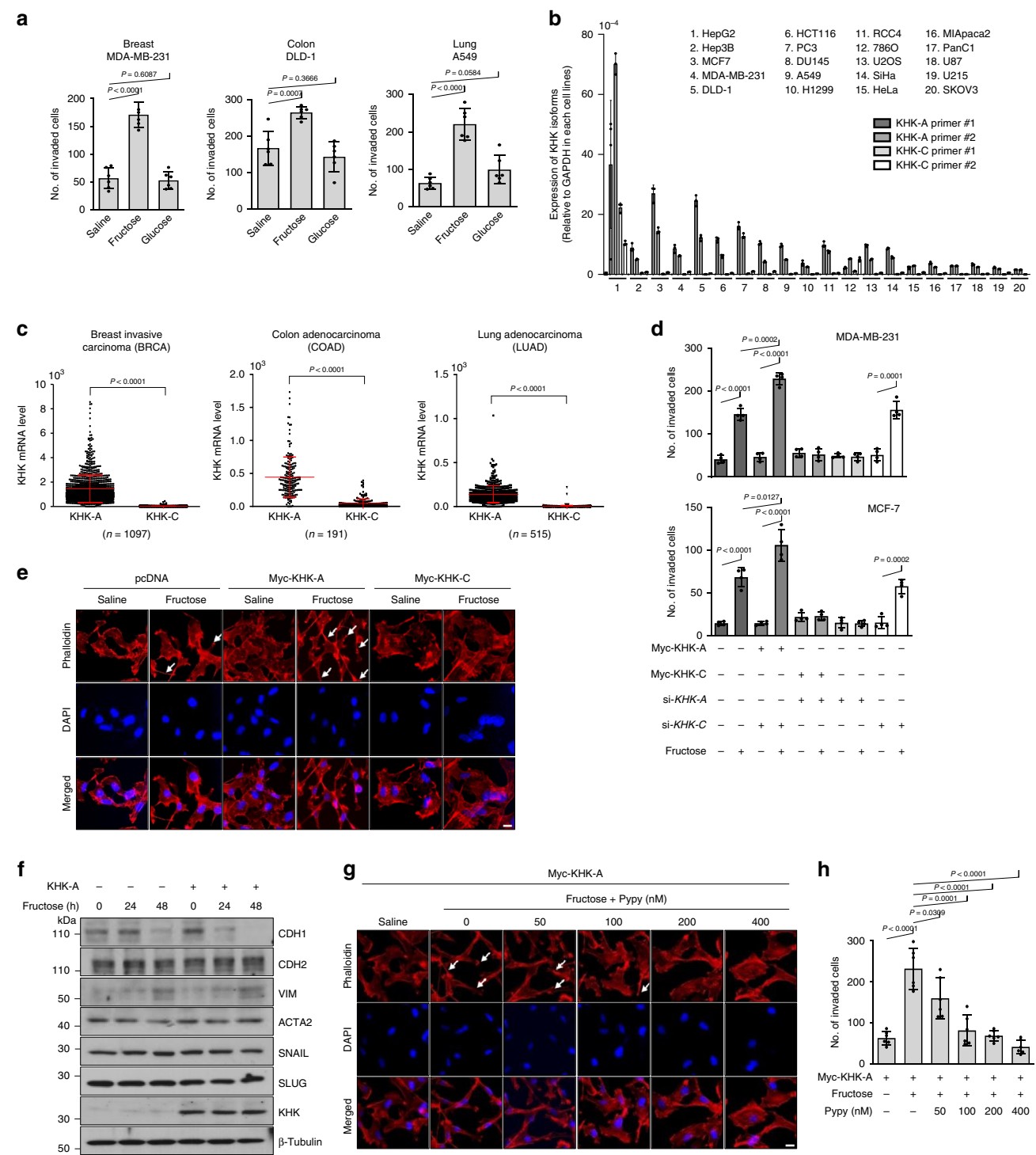

and Supplementary Fig. 6b. We found that the body weights of mice were severely reduced in group 'f,' (KHK-A-overexpressing tumor in fructose-fed mice) (Fig. 2b). Tumor growth was faster in three KHK-A-overexpressing groups than in three control groups. Fructose intake marginally enhanced tumor growth in the KHK-A groups, but not in the control groups (Fig. 2c). However, when endogenous KHK-A was silenced or replaced with ectopic KHK-C, fructose intake showed no more effects on tumor growth and body weight (Supplementary Fig. 6c, d). Because tumor volume can affect the chance of metastasis, the primary tumors were clearly resected when the tumor volume reached 500–600 mm³. Two weeks later, chest metastasis was detected in

six mice in group 'f,' whereas metastasis rarely developed in the other groups (Fig. 2d and Supplementary Figs. 5b and 6e). To further evaluate lung metastasis, bioluminescence emission was measured in excised lung tissues. In most lungs from group 'f,' bioluminescence was strongly emitted over entire areas (Fig. 2e). Weak emission was locally detected in two lungs from groups 'c' and 'h' (control tumors in fructose-fed mice, Fig. 2e and Supplementary Fig. 6f). Histological examination confirmed lung metastases in both fructose-feeding groups (Fig. 2f, and Supplementary Fig. 6g). In addition to the lung, tumor metastasis was detected in the liver, heart, hind leg, and GI tract in group 'f' (Fig. 2g). However, when KHK-A was silenced or replaced with

**Fig. 1 Fructose enhances the invasion potential of cancer cells. a** The invasion potential of cancer cells was analyzed using Boyden chamber with a matrigel-coated membrane. The indicated cells were treated with 5 mM fructose or glucose for 48 h and placed in the upper chamber. After 24 h, cells passing through the interface membrane were stained and counted. The numbers (means ± S.D. from six independent experiments) of invaded cells are presented as bar graphs. **b** The mRNA levels of KHK-A and KHK-C were measured by quantitative RT-PCR. The mRNA levels (means ± S.D. from three independent experiments) were normalized to the GAPDH level. **c** Isoform analysis in breast invasive carcinoma, colon, and lung adenocarcinoma data from TCGA. Data are presented as means ± S.D. from the number of samples derived from independent cancer patients were provided by TCGA database are shown in each panel. Significance was calculated by a two-sided Mann–Whitney $U$ test. **d** MDA-MB-231 and MCF-7 cells, which had been transfected with 1 μg of Myc-tagged plasmid or 60 nM siRNA, were subjected to the invasion assay (means ± S.D. from four independent experiments). **e** MDA-MB-231 cells, which had been transfected with Myc-KHK-A or -C, were treated with 5 mM fructose for 48 h. F-actin and nucleus were stained with Alexa Fluor 633 Phalloidine (red) and DAPI (blue), and visualized under a fluorescence microscope (scale bar = 10 μm). **f** MDA-MB-231 stable cells expressing KHK-A were incubated with 5 mM fructose, and subjected to western blotting for EMT markers. The blots are representative of three independent experiments. **g** MDA-MB-231 cells, which had neem transfected with Myc-KHK-A, were treated with 5 mM fructose and pyrimidinopyrimidine for 48 h. F-actin and nucleus were stained with Alexa Fluor 633 Phalloidine (red) and DAPI (blue), and visualized under a fluorescence microscope (scale bar = 10 μm). **h** Transfected MDA-MB-231 cells were treated with 5 mM fructose and pyrimidinopyrimidine, and subjected to Matrigel invasion assay (means ± S.D. from 4 independent experiments). In (**a**), (**d**), (**h**), significance was calculated by unpaired, two-sided Student's $t$-test. The results for significance tests are included in each panel.

KHK-C, fructose intake failed to promote tumor metastasis (Supplementary Fig. 6e–g). Collectively, fructose feeding promotes breast-cancer metastasis, which is exacerbated by KHK-A overexpression.

**Fructose facilitates KHK-A binding to LRRC59 and YWHAH.** Given that KHK-C has no effect on cell invasion, we hypothesized that KHK-A promotes cell invasion via some unique pathway irrespective of fructose metabolism. To identify the KHK-A pathway, we searched for proteins that interacted with KHK-A. In both MCF-7 and MDA-MB-231 cells, three proteins, glyoxalase domain-containing protein 4 (GLOD4), tyrosine 3-monooxygenase/tryptophan 5-monooxygenase activation protein Eta (YWHAH), and leucine-rich repeat-containing protein 59 (LRRC59), were commonly identified as interacting with KHK-A in a fructose-dependent manner (Fig. 3a). To determine which of the three proteins participate in KHK-A-promoted cell invasion, we knocked down each candidate. The fructose-induced invasions of MCF-7 and MDA-MB-231 were attenuated by silencing YWHAH or LRRC59 but not by silencing GLOD4 (Fig. 3b and Supplementary Fig. 7a). This result indicates that LRRC59 and YWHAH are involved in the KHK-A pathway for cell invasion. Immunoprecipitation analyses confirmed the fructose-dependent interaction of KHK-A, but not KHK-C, with the two proteins (Fig. 3c and Supplementary Fig. 7b, c). To examine where the proteins interact, we performed immunoprecipitation in subcellular fractions. KHK-A associated with LRRC59 in both the cytoplasm and the nucleus and with YWHAH in only the nucleus (Fig. 3d). Interestingly, the input results showed that fructose induced the nuclear translocation of KHK-A and LRRC59, which encouraged us to test the possibility that the nuclear transporter LRRC59 aids KHK-A with entering the nucleus.

**LRRC59 and KPNB1 transport KHK-A to the nucleus.** In the immunoblotting and immunofluorescence analyses, fructose-induced nuclear translocation of KHK-A in dose- and time-dependent manners (Fig. 4a, b and Supplementary Fig. 8a, c–e). By contrast, KHK-C stayed in the cytoplasm regardless of fructose treatment (Supplementary Fig. 8b–e). Compared with KHK-C, KHK-A has a higher $K_m$ value for fructose on the enzyme kinetics[18]. Based on this enzymatic property, KHK-A has been regarded to have a lower affinity to fructose, so it was questioned how fructose triggers the nuclear translocation of KHK-A. Therefore, we measured a physical affinity for KHK-A or KHK-C binding to radioactive fructose. In HEPES and phosphate buffers, the $K_d$ values (mM) of KHK-A for fructose were 0.53 and 0.55,

and those of KHK-C were 0.40 and 0.47, respectively (Supplementary Fig. 8f). These results suggest that KHK-A interacts with fructose as sensitively as KHK-C does but is ineffective in phosphorylating fructose. Given that the fructose-dependent nuclear translocation of KHK-A was attenuated by silencing LRRC59 (Fig. 4c), LRRC59 might function as a nuclear importer for KHK-A. An in vitro binding assay showed that the existence of fructose is a prerequisite for the association of KHK-A and LRRC59 (Fig. 4d). As LRRC59 is known to transport proteins in concert with KPNB1[23], we investigated the involvement of KPNB1 in the KHK-A translocation. Consequently, KPNB1 also joined to the fructose-dependent complex of LRRC59 and KHK-A (Fig. 4e and Supplementary Fig. 8g); the association of KPNB1 also depended on LRRC59 (Fig. 4f and Supplementary Fig. 8h). Given that the nuclear translocations of KHK-A and LRRC59 were blocked by KPNB1 knockdown (Fig. 4g and Supplementary Fig. 8i), KPNB1 is also required for fructose-dependent nuclear translocation. Next, we examined whether the Leu83 residue in the hydrophobic motif of KHK-A was essential for the LRRC59 interaction, as it was for the PRPS1 interaction[19]. The KHK-A_L83A mutant cannot interact with the nuclear importers (Fig. 4h and Supplementary Fig. 8j). Furthermore, KHK-A_L83A neither enters the nucleus, even under fructose stimulation (Fig. 4i, Supplementary Fig. 8k), nor promotes fructose-dependent cell invasion (Fig. 4j, Supplementary Fig. 8l). It should be noted that leucine 83 is present only in KHK-A because it is encoded by exon 3, which is spliced differently in KHK-A than in KHK-C. The isoform-specific exon 3 motif provides the unique action of KHK-A in response to fructose.

**KHK-A functions as a serine kinase for YWHAH.** First, we excluded the possibility that YWHAH participates in the fructose-induced nuclear translocation of KHK-A (Supplementary Fig. 9a). As the kinase KHK-A interacts with YWHAH in the nucleus, we hypothesized that KHK-A moonlights to phosphorylate YWHAH. An in vitro binding assay showed that KHK-A directly interacts with YWHAH (Fig. 5a). Surprisingly, KHK-A was found to induce Ser/Thr-phosphorylation of YWHAH (Fig. 5b). However, the kinase-dead mutant KHK-A_dATP, which lacked the ATP binding motif, failed to induce phosphorylation of YWHAH, even though it entered the nucleus in response to fructose (Fig. 5b and Supplementary Fig. 9b). In addition, Pypy also attenuated YWHAH phosphorylation (Supplementary Fig. 9c). These results further support our notion that YWHAH phosphorylation is mediated by the kinase reaction of KHK-A. An in vitro kinase assay confirmed that KHK-A directly phosphorylates YWHAH but KHK-C does not (Fig. 5c,

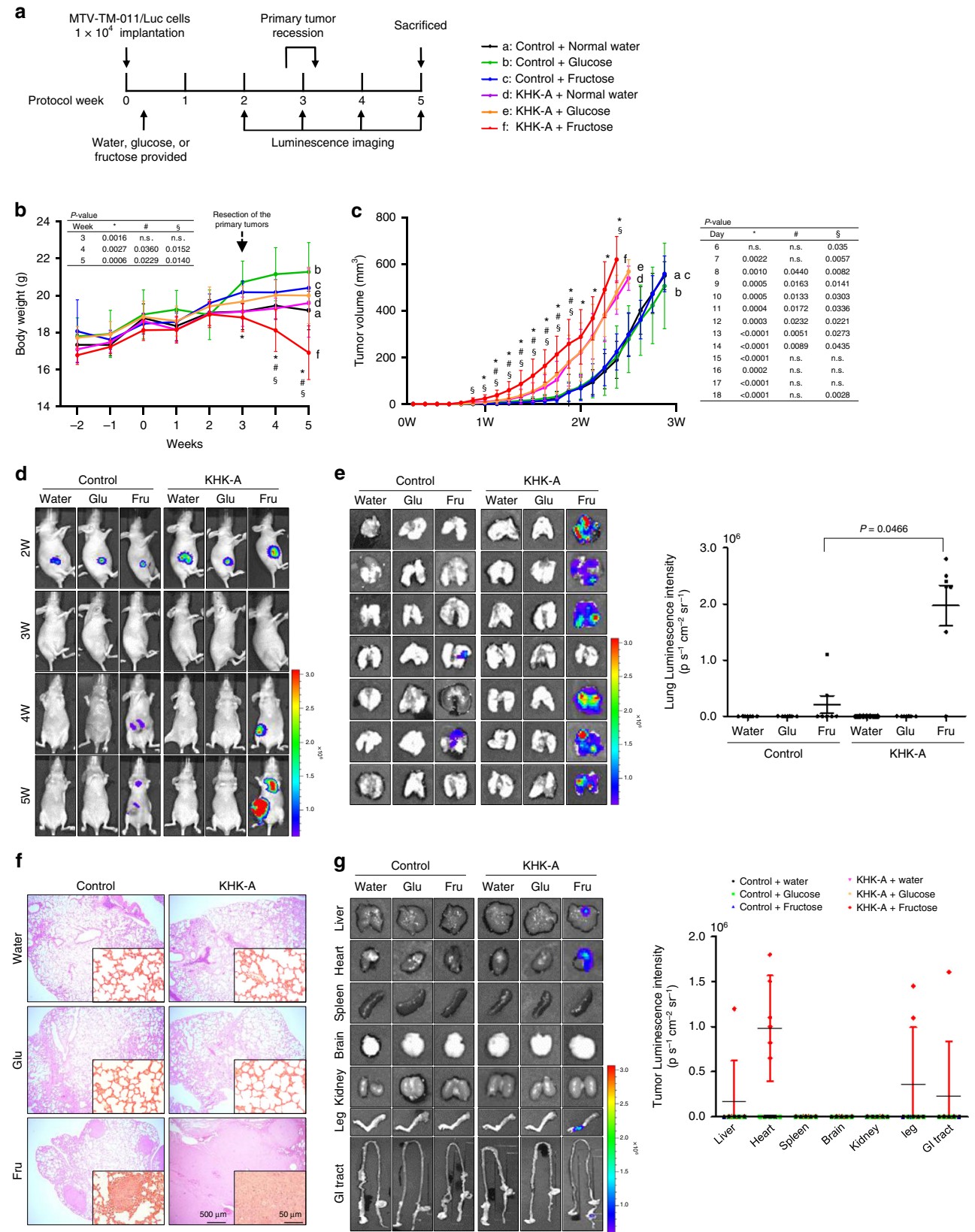

Supplementary Fig. 9d). Unexpectedly, the YWHAH phosphorylation was inhibited by co-incubation with 1 mM fructose. The fructose binding to the catalytic site of KHK-A seems to interfere with the KHK-A action on YWHAH phosphorylation. We further examined the effect of fructose on the KHK-A phosphorylation of YWHAH. In an enzyme kinetics analysis, the $K_m$ value of KHK-A for YWHAH was identified to be 246 nM (Supplementary Fig 9e). The $IC_{50}$ and $K_i$ values of fructose against the KHK-A phosphorylation of YWHAH were 0.91 and 0.72 mM, respectively (Supplementary Fig. 9f, g). To test the

**Fig. 2 KHK-A promotes breast-cancer metastasis in fructose-fed mice. a** Schematic diagram for the breast-cancer xenograft study. MTV-TM-011 stable cells expressing luciferase and KHK-A (or GFP) were implanted into mammary pads of mice. Tumor-bearing mice were fed with water, 15% glucose, or 15% fructose. Primary tumors were removed when the tumor volumes reached 500–600 mm³, and metastatic tumors were observed weekly by luminescence imaging. The condition of each experimental group (7 per each group) is described in the right panel. **b** Body weights of mice were checked once a week and expressed as the means ± S.D. from 7 mice per group. **c**. The volumes of breast tumors were monitored daily using calipers and expressed as the means ± S.D. from 7 mice per group. *Denotes $P < 0.05$ between group 'f' versus 'c'; # denotes $P < 0.05$ between group 'f' versus 'd'; § denotes $P < 0.05$ between group 'f' versus 'e'. **d** Bioluminescence images of tumor-bearing mice were taken weekly using Xenogen IVIS 100. The color bar represents bioluminescence intensity counts. **e** On the 5th week after cancer graft, the organs were excised from mice. Bioluminescence images were captured in the lungs (left). Bioluminescence intensity (photons/s/cm²/sr) in the lungs was quantitatively analyzed (right). Data are represented as means ± S.D. from the lungs harvested from 7 mice per group. **f** Representative pictures of H&E-stained lungs. **g** In other organs, representative bioluminescence images (left) and quantitative analyses of bioluminescence emission (right). Dots represent the bioluminescence intensities from individual samples, and horizontal bars show the means ± S.D. from all vital organs obtained from 7 mice per group. In (**b**), (**c**), (**e**), significance was calculated by a two-sided Mann–Whitney U test. The results for significance tests are included in each panel.

possibility that fructose inhibits the YWHAH kinase activity of KHK-A in the nucleus, we quantified the nuclear concentration of fructose using a gas-chromatography mass analysis. When cells were incubated with fructose, a substantial concentration of fructose was detected in the cytoplasmic fraction, but not in the nuclear fraction (Supplementary Fig 9h). These results suggest that the fructose inhibition of YWHAH phosphorylation does not occur in the nucleus.

To specify the site of YWHAH phosphorylation, we performed the Orbitrap tandem mass analysis, which indicates that KHK-A phosphorylates YWHAH at serine 25 in test tubes (Fig. 5d) and in cells (Supplementary Fig. 9i). To verify the S25 phosphorylation, we expressed the YWHAH_S25A mutant and found that the mutant was not more phosphorylated by KHK-A (Fig. 5e). This result indicates that Ser25 is the only residue for KHK-A-dependent phosphorylation. To evaluate the involvement of YWHAH in cell invasion, we checked the fructose-dependent F-actin rearrangement and found that YWHAH facilitates filopodia formation (Fig. 5f). Next, we removed endogenous YWHAH in MDA-MB-231 cells using siRNA targeting the 3′-UTR and then restored wild-type YWHAH or YWHAH_S25A. Consequently, YWHAH promoted the fructose-dependent invasion of human (Fig. 5g and Supplementary Fig. 9j) and murine (Supplementary Fig. 9k) breast-cancer cells, but the YWHAH_S25A mutant did not. Taken together, these data indicate that KHK-A mediates fructose-dependent cell invasion by phosphorylating YWHAH at S25.

Since PRPS1 and p62 were previously reported as substrates of KHK-A in liver cancer[19,24], we examined whether fructose facilitates the PRPS1 and p62 phosphorylation. Fructose did not affect the phosphorylation of PRPS1 in Hep3B, but slightly reduced that in MCF-7 and MDA-MB-231 (Supplementary Fig. 10a–c). As reported previously, p62 was found to be phosphorylated at Ser28 under hypoxia in liver and breast-cancer cells. HIF-1α was detected as a marker for hypoxia. Consequently, fructose did not induce the p62 phosphorylation (Supplementary Fig. 10d–f). Therefore, it is suggested that fructose does not enhance the protein kinase activity of KHK-A.

**Phosphorylation of YWHAH promotes breast-cancer metastasis.** For in vivo evaluation of breast-cancer metastasis, we established the MTV-TM-011 cell lines that stably co-express luciferase and YWHAH or YWHAH_S25A (Supplementary Fig. 11a). Mice (seven per group) were randomly allocated to six groups and subjected to tumor graft (Fig. 6a). The body weight was severely reduced in group 'd' (Fig. 6b). Tumor growth was faster in the YWHAH-overexpressing groups than the other groups (Fig. 6c). After the resection of primary tumors, metastasizing tumors were detected at the chests of seven mice in group 'd' and two mice in group 'b', whereas the metastasis rarely

developed in the other groups (Fig. 6d and Supplementary Fig. 11b). In group 'd', bioluminescence was strongly emitted over entire areas of the excised lungs, and weak emission was locally detected in several lungs from groups 'b' and 'f' (Fig. 6e). Histological examination confirmed metastatic tumor nodules in the lung tissues from fructose-fed mice (Fig. 6f). In group 'd', metastatic tumors were also detected in the livers, spleen, brain, and GI tracts (Fig. 6g). Given these results, the KHK-A-mediated phosphorylation of YWHAH is critical for the fructose-induced metastasis of breast cancer.

**Structural analysis for the KHK-A-mediated phosphorylation.** Considering that KHK-C phosphorylates a small molecule (fructose), it seems strange that KHK-A phosphorylates amino acid residues in proteins, such as PRPS1 and YWHAH. To clarify this conflict, we analyzed molecular dynamics (MD) simulation with an 11-residue peptide model for the local fragment of YWHAH around S25 (from Y20 to V30), namely YWHAH-11. The umbrella sampling technique[25,26] was used to compute the potential of mean force (PMF), which corresponds to the free energy change associated with YWHAH-11 experiencing its molecular environment, when the hydroxyl group at S25 of YWHAH-11 (S25-OH) approached the γ-phosphoryl group of ATP (ATP-γPO₃) on KHK-A. The PMF profile as a function of the distance ($d$) between the oxygen of S25-OH and the phosphorus of ATP-γPO₃ (Fig. 7a) indicates that YWHAH-11 can form a ternary complex with KHK-A-bound ATP by reaching a local minimum at $d \cong 3.5$ Å. The structure of the ternary complex YWHAH-11/ATP/KHK-A is shown in Fig. 7b. Such a P–O distance at the local minimum, which is the closest distance before the phosphorylation reaction occurs (i.e., reactant state), agrees well with previous quantum mechanical approaches[27,28] ($d = 3.2$–$3.3$ Å), which also identified the transition and product state of the phosphorylation reaction at $d = 2.2$–$2.3$ Å and $d = 1.7$–$1.8$ Å, respectively. Having obtained the PMF for the YWHAH-11/ATP/KHK-A complex, it is interesting to investigate how PMF can be altered for YWHAH/ATP/KHK-C and PRPS1/ATP/KHK-A. These systems were modeled by the same method, and their PMFs are compared in Fig. 7a. For the model of PRPS1/ATP/KHK-A, we employed the 11-residue peptide model, PRPS1-11, for the fragment of PRPS1 around T225 (from D219 to I229), which was similar to YWHAH-11. Notably, the PMF for PRPS1-11/ATP/KHK-A has a profile similar to that of YWHAH-11/ATP/KHK-A, with a nearly identical local minimum at $d \cong 3.6$ Å, whereas the profile for YWHAH-11/ATP/KHK-C has no minima in the entire profile. To further analyze this profile similarity in YWHAH-11/ATP/KHK-A and PRPS1-11/ATP/KHK-A, we computed the residue–residue contact maps for YWHAH-11/ATP/KHK-A and PRPS1-11/ATP/KHK-A at the local minimum state (Fig. 7c). In the contact map, the region of

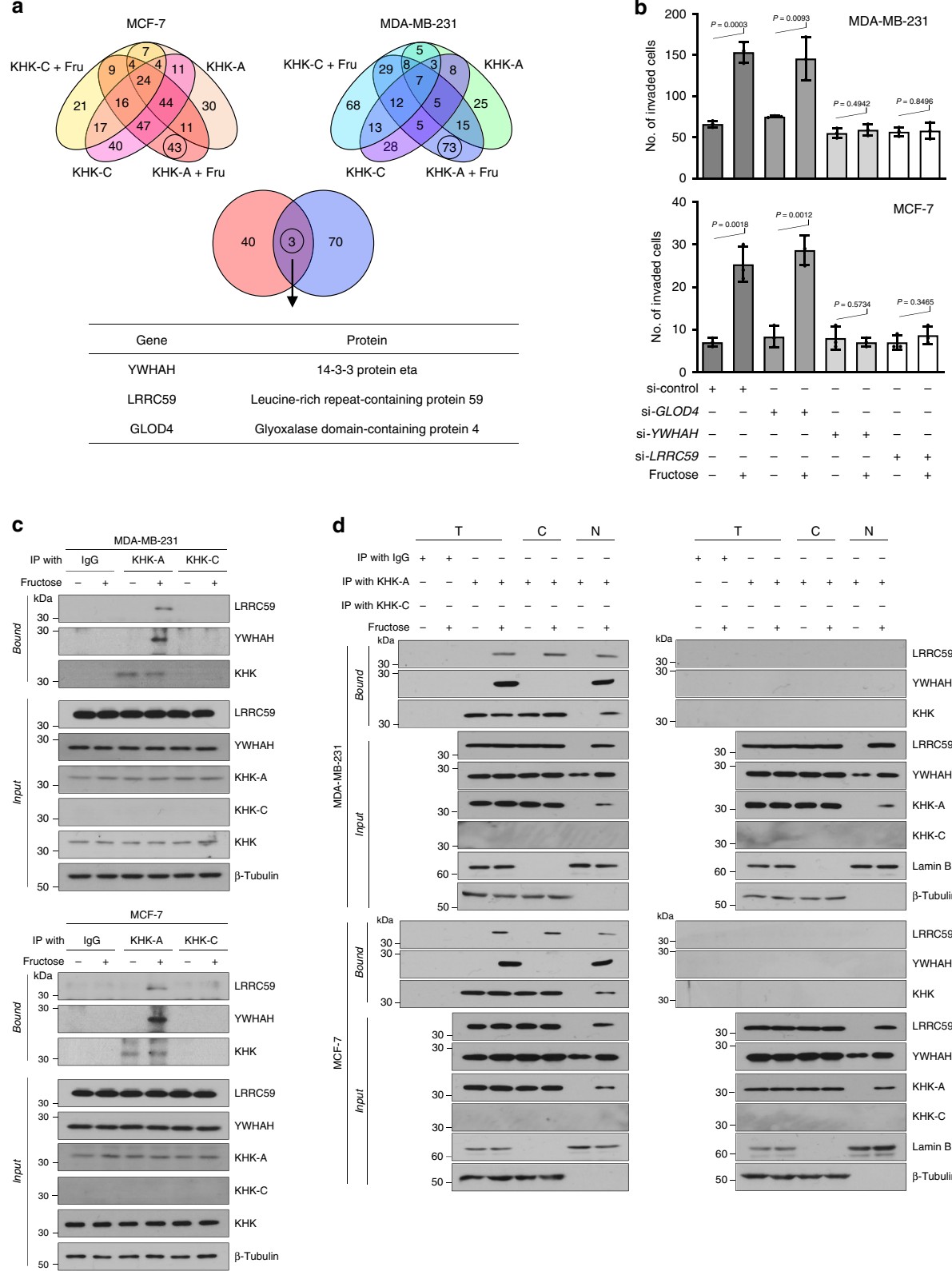

inter-chain contact between KHK-A (aa. 3–298) and YWHAH-11 or PRPS1-11 is specifically focused. Both the contact maps for KHK-A/YWHAH-11 and KHK-A/PRPS1-11 show similar inter-chain structure, revealing a feature that YWHAH-11 and PRPS1-11 commonly contact with a KHK-A region (aa. 100–114), which lies in the exon 3-derived segment differing from that of KHK-C[29]. These results demonstrate that the KHK-A region plays an

important role in the phosphorylation of YWHAH and PRPS1, presumably acting as an auxiliary binder for holding YWHAH and PRPS1 near ATP site. In particular, the average numbers of atoms in contact with the binding region of KHK-A are very high for D21 and D22 in YWHAH (top in Fig. 7d) and for D220 and D221 in PRPS1 (bottom in Fig. 7d). Interestingly, the amino acids 'DDMA' are commonly placed just before the KHK-A target

**Fig. 3 Identification of proteins interacting with KHK-A fructose-dependently. a** Proteomics analyses to identify KHK-A-specific interactome. MCF-7 and MDA-MB-231 cells, which had been transfected with His-KHK-A or -C, were incubated with or without 5 mM fructose for 8 h. His-tagged proteins were pulled down using Ni-affinity beads, eluted by imidazole, and analyzed by LC-MS/MS. This experiment was performed three times independently. Venn diagrams show the numbers of identified proteins in each group (the top panels). Three KHK-A-interacting proteins were identified commonly in two cell lines and are listed in the bottom panel. **b** MCF-7 and MDA-MB-231 cells were transfected with the indicated siRNAs, incubated with 5 mM fructose for 48 h, and subjected to Matrigel invasion analysis. The numbers (means ± S.D. from three independent experiments) of invaded cells are presented as bar graphs. Significance was calculated by unpaired, two-sided Student's *t*-test. **c** After cells were incubated with 5 mM fructose for 8 h, and cell lysates were immunoprecipitated with IgG, anti-KHK-A, or anti-KHK-C, and immunoblotted with anti-LRRC59 or anti-YWHAH antibody. **d** Subcellular localization of KHK-interacting proteins. After cells were incubated with 5 mM fructose for 8 h, total lysates (T) were fractionated to cytosolic (C) and nuclear (N) components, which were immunoprecipitated with IgG, anti-KHK-A, or anti-KHK-C, and immunoblotted with the indicated antibodies.

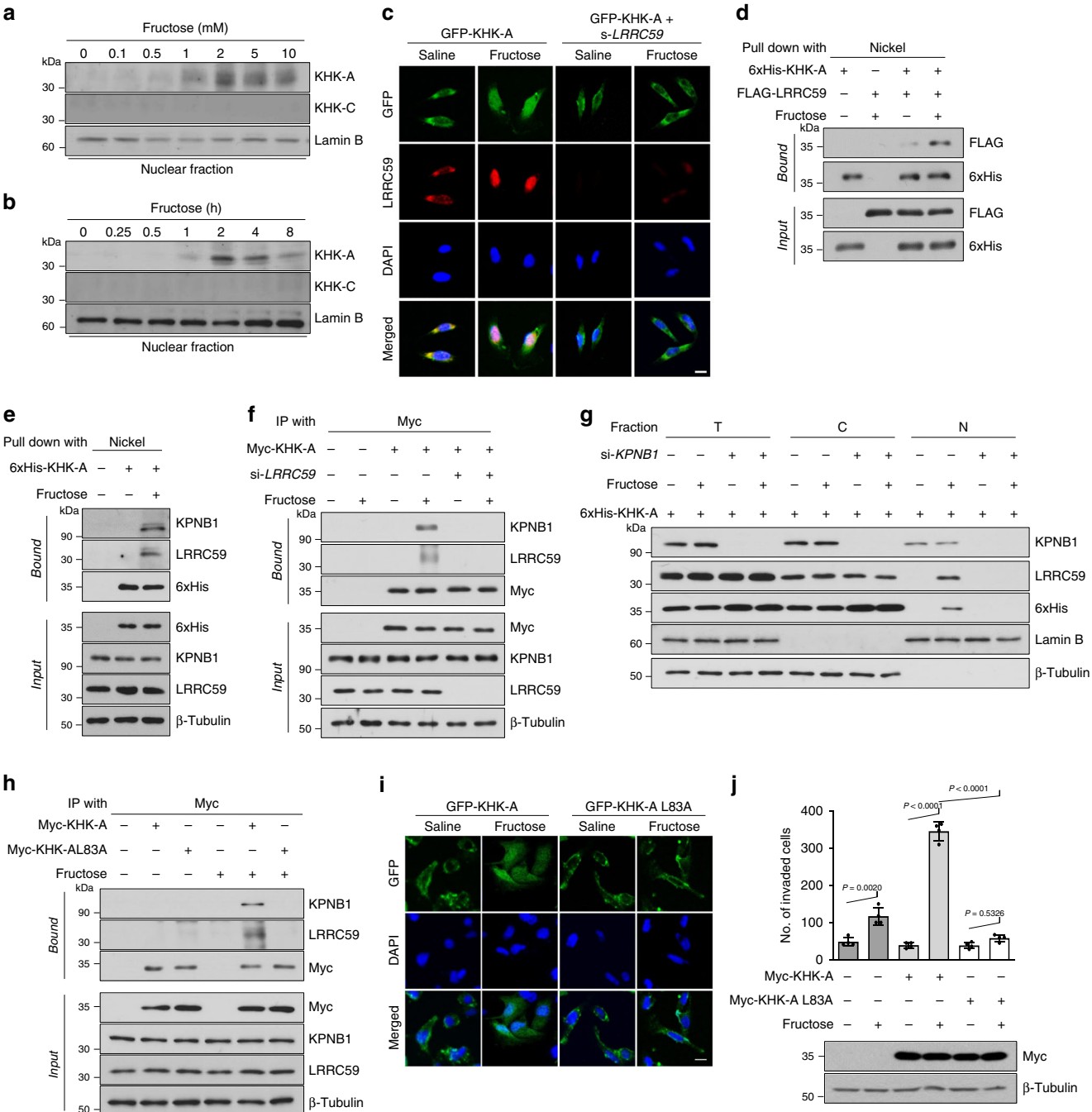

**Fig. 4 LRRC59 in concert with KPNB1 mediates the nuclear translocation of KHK-A. a** MDA-MB-231 cells were treated with fructose at the indicated concentrations for 2 h. KHK-A and KHK-C were immunoblotted in the nuclear fraction. **b** MDA-MB-231 cells were treated with 5 mM fructose for the indicated times. Nuclear KHK-A and KHK-C were immunoblotted. **c** Immunofluorescence images of GFP-tagged KHK-A (green), endogenous LRRC59 (red), and DAPI (blue). Scale bar = 10 μm. **d** In vitro binding of KHK-A and LRRC59. His-KHK-A or Flag-LRRC59 proteins were purified from HEK293T cells using Nickel-NTA or anti-Flag antibody, and incubated together for 1 h with or without 1 mM fructose. His-KHK-A was pulled down using Nickel-NTA and immunoblotted with anti-Flag antibody. **e** KHK-A binding to LRRC59 and KPNB1. MDA-MB-231 cells expressing His-KHK-A were treated with fructose. His-KHK was pulled down using Nickel-NTA, and the co-precipitated LRRC59 and KPNB1were immunoblotted. **f** LRRC59 is required for the KHK-A binding to KPNB1. MDA-MB-231 cells, which had been co-transfected with Myc-KHK-A and LRRC59 siRNA, were incubated with fructose. Myc-KHK-A was immunoprecipitated and co-precipitated proteins were immunoblotted. **g** KPNB1-dependent nuclear translocation of KHK-A. MDA-MB-231 cells, which had been co-transfected with His-KHK-A and KPNB1 siRNA (or control siRNA), were treated with fructose. The cell fractions were immunoblotted. **h** The L83 residue participates in the KHK-A interaction with LRRC59 and KPNB1. MDA-MB-231 cells were transfected with Myc-KHK-A or Myc-KHK-A_L83A, and incubated with fructose. The cell lysates were immunoprecipitated with anti-Myc and immunoblotted with anti-LRRC59 or anti-KPNB1. **i** Immunofluorescence imaging of GFP-tagged KHK-A wild type or L83A (green) in MDA-MB-231 cells which were treated with fructose. All samples were counterstained with DAPI (blue). Scale bar = 10 μm. **j** MDA-MB-231 cells were transfected with Myc-KHK-A wild-type or L83A mutant plasmid, and treated with 5 mM fructose for 48 h, then subjected to invasion assay. The numbers (means ± S.D. from four independent experiments, *P*-value was calculated by unpaired, two-sided Student's *t*-test) of invaded cells shown in the bar graphs. In (**c**), (**e**), (**f**), (**g**), (**h**), (**i**), transfected MDA-MB-231 cells were incubated with 5 mM Fructose for 8 h.

residues in YWHAH and PRPS1. Our MD simulation results also revealed that the double aspartate residues in 'DDMA' play a crucial role in navigation of YWHAH or PRPS1 toward ATP bound to KHK-A. This was verified by the PMF profile computed for YWHAH-11 mutated with D21A (YWHAH-11_D21A), which clearly shows a shifting of distance *d* at the local minimum to a larger value (Fig. 7a). With this information on MD simulation, we experimentally tested the possibility that the D21 residue is essential to the KHK-A phosphorylation of YWHAH. The YWHAH_D21A mutant neither interacted with KHK-A nor phosphorylated it (Fig. 7e, f). Taken together, these data further support our notion that KHK-A can phosphorylate the S25 of YWHAH or the T225 of PRPS1.

**S25-phosphorylated YWHAH recruits SLUG to the CDH1 promoter**. Among representative EMT markers, CDH1 expression was the most strikingly changed by fructose in breast-cancer cell lines. The fructose-dependent suppression of CDH1 was augmented by KHK-A overexpression, which was reversed by YWHAH knockdown (Fig. 8a and Supplementary Fig. 12a, b). CDH1 expression was regulated at the transcriptional level (Fig. 8b and Supplementary Fig. 12c, d). A previous study demonstrated that YWHAH regulates CDH1 expression by interacting with the transcription factor SNAIL[30]; therefore, we examined whether YWHAH interacts with SNAIL and other CDH1 repressors. As reported, YWHAH interacted with SNAIL, but this interaction was independent of fructose or KHK-A. By contrast, SLUG was found to interact with YWHAH in fructose- and KHK-A-dependent manners (Fig. 8c, and Supplementary Fig. 12e, f). In chromatin immunoprecipitation analyses, the recruitments of YWHAH and SLUG to the *CDH1* promoter were shown to depend on fructose and KHK-A (Fig. 8d, e, Supplementary Fig. 12h,i), but the SNAIL recruitment was not (Supplementary Fig. 12g). Despite its existence in the nucleus (Supplementary Fig. 12j), YWHAH_S25A did not interact with SLUG in response to fructose (Fig. 8f, Supplementary Fig. 12k). This indicates that the S25 phosphorylation of YWHAH is critical for the SLUG interaction. As expected, the fructose-dependent suppression of CDH1 was diminished by SLUG knockdown in the presence of YWHAH but not in the presence of YWHAH_S25A (Fig. 8g and Supplementary Fig. 12l). These results were also verified at the mRNA level of *CDH1* (Fig. 8h and Supplementary Fig. 12m, n). We next examined whether the fructose-dependent phosphorylation of S25 residue was essential for the recruitment of YWHAH to the *CDH1* promoter. The binding of YWHAH to the *CDH1* promoter was enhanced by

fructose, whereas that of YWHAH_S25A was not affected by fructose (Fig. 8i). Likewise, fructose facilitated SLUG binding to the *CDH1* promoter, which was also augmented by wild-type YWHAH (Fig. 8j). As expected, the binding of SNAIL to the *CDH1* promoter was not regulated by S25 phosphorylation (Supplementary Fig. 12o). We assessed whether the sequential binding of proteins also affected cell invasiveness. An invasion assay was performed on MDA-MB-231 or MTV-TM-011 cells that were stably expressing KHK-A. Fructose-induced invasiveness was determined by SLUG in YWHAH-expressing cells but not in YWHAH_S25A-expressing cells (Fig. 8k and Supplementary Fig. 12p, q). To examine the degree to which the CDH1 suppression contributes to fructose-induced invasion, we restored CDH1 expression in KHK-A-overexpressing cells, which almost completely blocked the fructose-induced invasion (Fig. 8l and Supplementary Fig. 12r, s). This result suggests that CDH1 suppression is the key event for cell invasion. Finally, we examined the KHK-A signaling pathway in breast-cancer tissues, which were obtained from mice presented in Fig. 2, and Supplementary Fig. 5. Immunohistochemical analyses showed the suppression of CDH1 and the nuclear expressions of KHK and LRRC59 in the tumors of fructose-fed mice (Fig. 8m and Supplementary Fig. 13a, b). We also confirmed the YWHAH phosphorylation in the nucleus by immunostaining breast-cancer grafts that were primarily removed from mice in Fig. 6 (Supplementary Fig. 13c). Taken together, these data suggest that fructose induces cancer invasion through the KHK-A-YWHAH-SLUG-CDH1 pathway.

**KHK-A-YWHAH-pSer25 axis correlates with metastasis**. Next, we investigated whether the KHK-A phosphorylation of YWHAH is clinically associated with breast-cancer metastasis. To examine the association of KHK-A and S25-phosphorylated YWHAH, we immunologically stained human breast-cancer arrays (Fig. 9a). The cancer specimens were histologically graded according to the Nottingham grading system (Fig. 9b). The nuclear KHK-A and S25-phosphorylated YWHAH levels both were significantly higher in the grade 3 group than in the grade 2 group (Fig. 9c). Even when the breast-cancer specimens were divided into non-metastasis and metastasis groups, the nuclear KHK-A and S25-phosphorylated YWHAH levels were much higher in the metastasis group (Fig. 9d). Spearman correlation analysis showed that nuclear KHK-A expression correlates with the S25-phosphorylated YWHAH expression (Fig. 9e). Collectively, the KHK-A-mediated phosphorylation of YWHAH is likely to be clinically associated with breast-cancer metastasis.

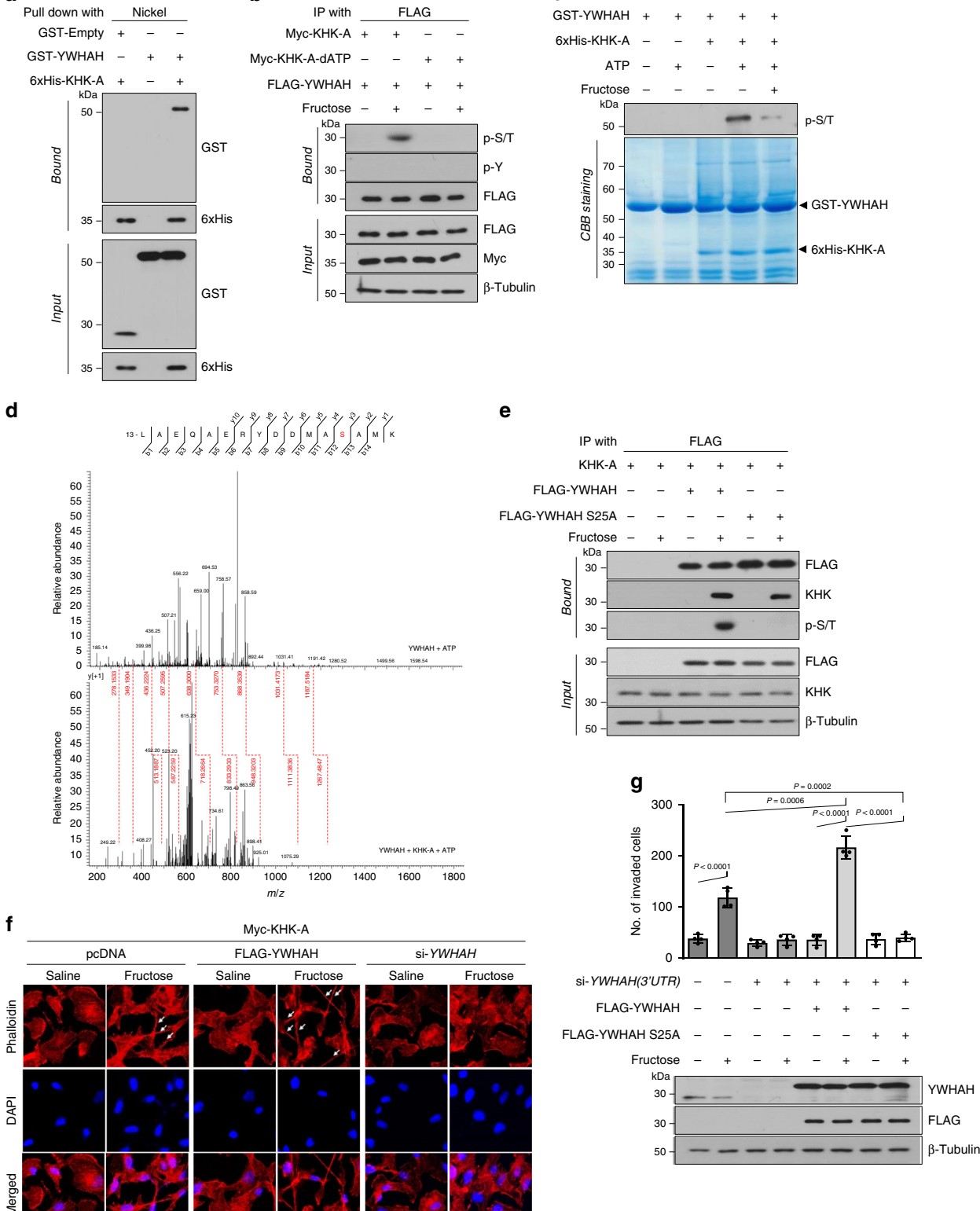

## Discussion

To explain how fructose promotes cancer development and progression, many investigators have focused on KHK-C-initiated fructose flux in energy metabolism. Fructose metabolism seems to provide cancer cells with the supplementary fuel required for proliferation and metastasis in colon cancer, and glioma[11–13]. According to our results, however, most cancer cell lines predominantly express KHK-A rather than KHK-C, and KHK-A mediates the fructose-induced cell invasion. As KHK-A does little to phosphorylate fructose at physiological concentrations due to its high $K_m$ value[18], it has remained somewhat mysterious that fructose promotes cancer progression through metabolic reprogramming in cancer cells lacking KHK-C[14–16]. Herein, we demonstrated a protein kinase function of nuclear

**Fig. 5 KHK-A phosphorylates YWHAH at Ser25. a** In vitro binding assay. Purified recombinant GST-YWHAH (or GST) were incubated with purified KHK-A in a test tube for 1 h and pulled down using nickel-affinity beads. Interaction between KHK-A and YWHAH was evaluated by immunoblotting. **b** KHK-A-dependent phosphorylation of YWHAH. FLAG-YWHAH was co-transfected with Myc-KHK-A or Myc-KHK-A-dATP. After 8-h incubation with 5 mM fructose, MDA-MB-231 cells were subjected to immunoprecipitation and immunoblotting. **c** In vitro kinase assay. Recombinant GST-YWHAH and His-KHK-A were co-incubated without or with 1 mM fructose in a kinase buffer. The Ser/Thr-phosphorylation of YWHAH was evaluated by immunoblotting, and the protein loading was verified by Coomassie staining. **d** LC-MS/MS analysis. Recombinant GST-YWHAH was reacted with or without GST-KHK-A in a kinase buffer, electrophoresed on SDS-PAGE, and digested in gel. The phosphorylation of Ser25 residue in YWHAH was identified based on LC-MS/MS spectra. **e** MDA-MB-231 stable cells expressing KHK-A, which had been co-transfected with Flag-YWHAH or S25A plasmid, were incubated with 5 mM fructose for 8 h. The cell lysates were immunoprecipitated with anti-FLAG, and the phosphorylation of YWHAH was evaluated by immunoblotting. **f** MDA-MB-231 cells, which had been co-transfected with Myc-KHK-A and si-YWHAH, were incubated with 5 mM fructose for 48 h. F-actin and nucleus were stained with Alexa Fluor 633 Phalloidine (red) and DAPI (blue), and visualized under a fluorescence microscope (scale bar = 10 μm). White arrows indicate filopodia. **g** Endogenous YWHAH was silenced in MDA-MB-231 cells using an siRNA targeting the 3′-UTR region of YWHAH mRNA, and YWHAH or S25A mutant proteins were restored in the cells. After incubated with 5 mM fructose for 48 h, cells were subjected to Matrigel invasion assay. The numbers (means ± S.D. from four independent experiments) of invaded cells are shown as bar graphs. Significance was calculated by unpaired, two-sided Student's t-test.

KHK-A in fructose-induced metastasis of breast cancer. Upon fructose stimulation, KHK-A is transported to the nucleus by the importers LRRC59 and KPNB1, and there phosphorylates YWHAH at S25. The phospho-YWHAH downregulates CDH1 by recruiting SLUG to the CDH1 promoter, thereby facilitating cell migration. In conclusion, fructose cues KHK-A to start a cell movement-promoting job. The signaling pathway for fructose-induced metastasis is summarized in Fig. 10.

The KHK-C-driven fructose metabolism has been intensively investigated as a pathologic pathway that aggravates metabolic disorders with excessive intake of fructose[2–4]. The harmful effect of the fructose metabolism has been also examined in cancer progression[11]. On the contrary, KHK-A is known to make a minor contribution to fructose-mediated metabolic disorders. Interestingly, a recent report suggested a negative function of KHK-A in fructose-mediacted metabolic disorders[31]. The KHK-A knock-out aggravated the metabolic profile in AldoB knock-out mice, suggesting that KHK-A plays a protective role in the fructose-mediated metabolic disorders. Yet, the mechanism underlying such a function of KHK-A has not been understood. Given two previous reports[19,24] and our data, KHK-A might be a protein kinase responsible for cell growth, stress response, and motility. Therefore, it is speculated that KHK-A fine-tunes the fructose-driven metabolic reprogramming and this function is related to the protein kinase activity of KHK-A.

Although KHK-C and KHK-A are very similar in terms of molecular structures, they phosphorylate distinct substrates: a small metabolite (fructose) and a polypeptide (YWHAH), respectively. The role of KHK-A as a protein kinase is consistent with a previous report showing that KHK-A phosphorylates PRPS1. KHK-A phosphorylates PRPS1 at Thr225 in the cytoplasm and activates it, which stimulates de novo nucleic acid synthesis through the pentose phosphate pathway; by so doing, it promotes hepatoma formation[19]. Taken together, our data indicate that KHK-A seems to be intrinsically a Ser/Thr protein kinase rather than a fructose kinase. Then, is there any conserved domain targeted by KHK-A? Comparing the amino acid sequences around the phosphorylated Ser/Thr residues, the sequence 'DDMA' commonly precedes the phosphorylated residues in YWHAH and PRPS1. Using molecular dynamics analysis, we suggested that two Ds in the conserved motif play a key role in the substrate binding to KHK-A, which was also experimentally supported. The DDMA motif may be a landmark for identifying new protein substrates targeted by KHK-A and warrants further investigation.

LRRC59 was originally considered a ribosome receptor located at the endoplasmic reticulum, but its biological functions were not fully understood. Rather than a ribosomal receptor, LRRC59 is now considered a nuclear translocator for the growth factor FGF1 endocytosed by cells, which occurs in concert with KPNB1[23]. We found that the nuclear translocation of KHK-A is fructose-dependently driven by the LRRC59-KPNB1 complex. In this process, fructose plays an essential role in forming the trimeric complex of KHK-A, LRRC59, and KPNB1. The precise role of fructose in this interaction remains an open question. It is speculated that fructose may change the conformation of KHK-A to a structure favorable to interaction with LRRC59 and KPNB1.

The YWHA (or 14-3-3) family is composed of seven members (β, γ, ε, σ, ζ, τ, and η) and is widely expressed in all eukaryotic cells. The family members are phosphorylated by their upstream kinases and then bind to the phospho-serine/threonine motifs of their downstream effectors. Therefore, they are regarded as the adapters that mediate the protein kinase signaling pathways. As part of apoptosis, c-Jun N-terminal kinase (JNK) phosphorylates 14-3-3ζ and β at Ser186 or 14-3-3σ at Ser184. This phosphorylation dissociates the 14-3-3 proteins from BAX, and the free BAX then enters the mitochondrion to induce apoptosis by releasing cytochrome c[32]. Under GM-CSF stimulation, Src kinase phosphorylates 14-3-3ζ at Tyr179, and the phospho-14-3-3ζ activates phosphoinositide 3-kinase (PI3K) to increase the survival potential[33]. Aside from the aforementioned functions, the 14-3-3 proteins are believed to be involved in diverse cellular processes. In this work, we identified a phosphorylation site (Ser25) in 14-3-3η (YWHAH). Surprisingly, the upstream kinase for 14-3-3η is a metabolic enzyme, KHK-A, and the downstream effector of 14-3-3η is the transcription repressor SLUG. Among many functions, 14-3-3 proteins may act as transcriptional adapters to modulate gene expression.

Within the YWHA family, five members have been reported to interact with the RXXXpS/TXP motifs in the SNAG and ZF (zinc finger) domains of SNAIL, and recruit SNAIL to the E-box element in the CDH1 promoter[30]. SLUG, which is alternatively named SNAIL2, also has two putative 14-3-3 binding motifs, which are highly homologous with the RXXXpS/TXP motifs. As expected, we found that SLUG interacts with YWHAH. Interestingly, it is only after being phosphorylated at Ser25 by KHK-A that YWHAH acquires binding affinity to SLUG. As a consequence, YWHAH guides SLUG to gain access to the CDH1 promoter. By contrast, SNAIL and YWHAH were found to be constantly associated regardless of fructose stimulation or the phosphorylation of YWHAH. Given these results, it is suggested that SNAIL controls the basal level of CDH1, but SLUG takes part in the dynamic regulation of CDH1 expression in response to fructose stimulation. The molecular dynamics must be further investigated to elucidate how S25 phosphorylation permits YWHAH to interact with SLUG.

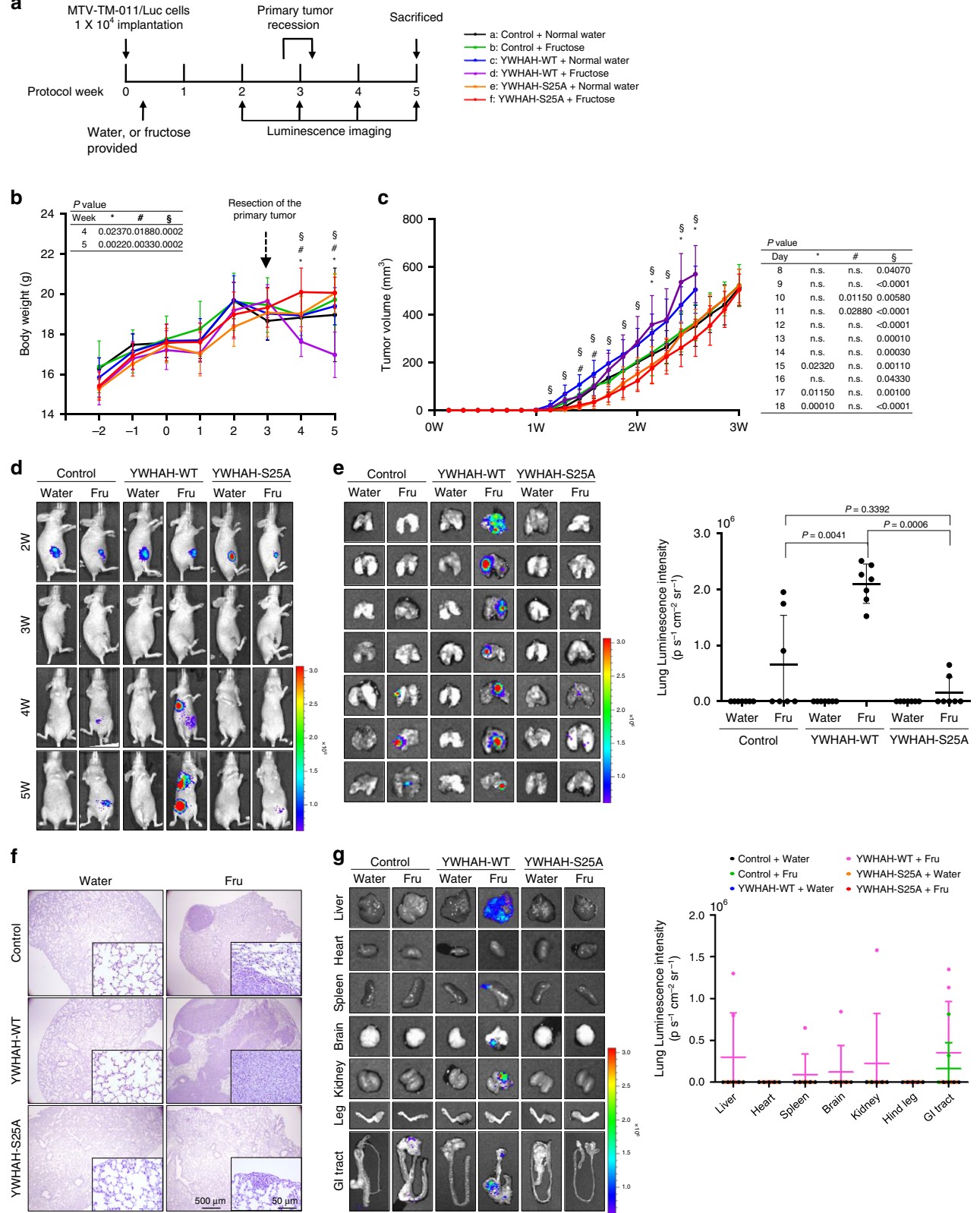

In conclusion, we observed that KHK-A promotes breast-cancer metastasis upon fructose stimulation and also demonstrated a signaling pathway responsible for fructose-induced metastasis. This work provides a reasonable mechanism supporting the clinical evidence for fructose-induced cancer aggravation. Based on our results, high intake of fructose should be restricted in cancer patients to reduce the risk of metastasis. From a therapeutic perspective, the KHK-A signaling pathway could be a potential target to prevent cancer metastasis.

## Methods

**Cell lines and cell culture.** Human breast cancer (MDA-MB-231 and MCF-7), embryonic kidney (HEK293T), glioblastoma (U87 and U251), lung adenocarcinoma (H1299 and A549), colon cancer (HCT116 and DLD-1), renal cancer (786O

**Fig. 6 Phosphorylated YWHAH promotes breast-cancer metastasis in fructose-fed mice. a** Schematic diagram for the breast-cancer xenograft study. The MTV-TM-011 stable cell line co-expressing luciferase and YWHAH or YWHAH-S25A were implanted into the mammary pads of mice. Tumor-bearing mice were fed with water or 15% fructose. After primary tumors were removed, metastatic tumors were checked weekly by luminescence imaging. The condition of each experimental group (7 per each group) is described in the right panel. **b** Body weights of mice (the means ± S.D. from 7 mice per group). **c** The volumes of breast tumors (the means ± S.D. from 7 mice per group). *Denotes $P < 0.05$ between group 'd' versus 'b'; # denotes $P < 0.05$ between group 'd' versus 'c'; § denotes $P < 0.05$ between group 'd' versus 'f'. **d** Bioluminescence images of tumor-bearing mice. **e** On the 5th week after cancer graft, bioluminescence images were captured in the excised lungs (left). Bioluminescence intensity (photons/s/cm$^2$/sr) was quantitatively analyzed (right). Data are represented as means ± S.D. from the lungs from 7 mice per group. **f** Representative pictures of H&E-stained lungs. **g** In other organs, representative bioluminescence images (left) and quantitative analyses of bioluminescence emission (right). Dots represent the bioluminescence intensities from individual samples, and horizontal bars show the means ± S.D. from all vital organs obtained from 7 mice per group. In (**b**), (**c**), (**e**), a two-sided Mann–Whitney $U$ test was used to calculate significance. $P$-values are represented in each panel.

and RCC4), cervical cancer (SiHa and HeLa), ovarian cancer (SKOV3), bone cancer (U2OS), and hepatocellular carcinoma (HepG2 and Hep3B) cell lines were obtained from American Type Culture Collection (ATCC; Manassas, VA); human pancreatic cancer (MIApaca2 and PanC1), human prostate cancer (PC3 and DU145), and mouse breast-cancer (MTV-TM-011) cell lines from Korean Cell Line Bank (Seoul, Korea). MDA-MB-231, MCF-7, MTV-TM-011, U251, H1299, A549, DU145, PC3, 786O, RCC4, HeLa, SKOV3, U2OS, and Hep3B were cultured in RPMI1640; U87 and HepG2 in MEM; HEK293T, MIApaca2, PanC1, DLD-1, HCT116, and SiHa in DMEM, which were supplemented with 10% heat-inactivated FBS. Mycoplasma contamination was routinely checked when cell growth or shape was altered. After thawing, cells were usually cultured for no more than 3 months. Cells were grown in a humidified atmosphere containing 5% CO$_2$ at 37 °C. Luciferase-expressing, luciferase/KHK-A, and luciferase/KHK-C-co-expressing MCF-7, MDA-MB-231, and MTV-TM-011 stable cell lines were established from G418-resistant clones for each one.

**Antibodies and reagents**. A ketohexokinase inhibitor Pypy (420640) was purchased from Millipore (Burlington, MA). Culture media, FBS, D-fructose (F3510), D-glucose (G7021), and anti-FLAG antibody (F7425, 1:5000 dilution) were purchased from Sigma-Aldrich (St. Louis, MO). Antibodies against KHK (sc-377411, 1:500), Slug (sc-166476, 1:1000), N-cadherin (sc-7939, 1:1000), Vimentin (sc-7558, 1:1000), ZEB-1 (sc-25388, 1:1000), β-tubulin (sc-9104, 1:5000), GST-tag (sc-138, 1:1000), PRPS1/2 (sc100288, 1:1000), Aldolase B (sc393278, 1:1000), and Lamin-B (sc-6216, 1:1000) were purchased from Santa Cruz Biotechnology (Santa Cruz, CA); antibodies against Importin-β (ab2811, 1:1000), α-SMA (ab7817, 1:1000), TWIST (ab50581, 1:1000), and Snail (ab53519, 1:1000) from Abcam (Cambridge, UK); antibodies against YWHAH (9640, 1:1000), Myc-tag (2278, 1:1000), GFP (2555,1:1000), and SQSTM1/p62 (5114T, 1:1000) from Cell Signaling Technology (Danvers, MA); antibodies against phosphor-serine/threonine (PP2551, 1:1000) and phosphor-tyrosine (PP2221, 1:1000) from ECM biosciences (Versailles, KY); antibodies against LRRC59 (NBP1-93953, 1:1000) and 12-lipoxygenase (NBP1-90338, 1:1000) were from Novus Biologicals (Littleton, CO); isotype-specific antibodies against KHK-A (21708-2, 1:1000) and KHK-C (21709-2, 1:1000) from Signalway Antibody LLC (Pearland, TX, USA); antibodies against His(6)-tag (PM032, 1:1000) from MBL (Nagoya, Japan). HRP-conjugated rabbit anti-goat (81-1620, 1:5000), anti-E cadherin (131700, 1:1000), and SQSTM1/p62 phospho-Ser28 (PA5-35409, 1:1000) antibodies from Thermo Fisher Scientific (Waltham, MA); HRP-conjugated goat anti-rabbit (G21234, 1:5000) and HRP-conjugated goat anti-mouse (G21040, 1:5000) from Invitrogen (Carlsbad, CA); anti-GLOD4 (GTX104484, 1:1000) from Genetex (San Antonio, TX). Anti-HIF-1α antibody (1:1000) was raised in rabbits against the ODDD peptide of human HIF-1α[34]. A monoclonal antibody against S25-phosphorylated YWHAH was raised using a phage display technology through a commercial facility (Bioneer, Daejeon, South Korea). The synthetic peptides (RYDDMASAMKAVTE and RYDDMA(p)SAM-KAVTE) were purchased from GL Biochem (Shanghai, China) and conjugated with bovine serum albumin to be used as antigens.

**Preparation of plasmids, siRNAs, and transfection**. The cDNAs for human KHK-A (NM_000221), human KHK-C (NM_006488), human LRRC59 (NM_018509), and human YWHAH (also known as 14-3-3η; NM_003405) were cloned by RT-PCR using Pfu DNA polymerase and inserted into pcDNA-Myc, pcDNA-FLAG, pcDNA-GFP, pcDNA-His(6), CMV luciferase/IRES, or pGEX plasmids using blunt-end ligation. The mutations of KHK-A and YWHAH were performed using PCR-based mutagenesis. The kinase-defective KHK-A (KHK-A-dATP) was constructed by deleting a.a. 255–260. The E-cadherin plasmid was purchased from Addgene (#47502; Cambridge, MA). For transient transfection of plasmids or siRNAs, cells were ~70% confluence were transfected using Lipo-fectamine 3000 (for plasmid) or Lipofectamine RNAiMAX (for siRNA), respectively. The transfected cells were stabilized for 48 h before being used in experiments. Nucleotide sequences of siRNAs are summarized in Supplementary Table 1.

**Establishment of stable cell lines with lentiviral vectors**. Lentiviral vectors containing mouse Khk-a-silencing shRNAs were generated by GenePharma (Shanghai, China). Oligonucleotides were annealed and inserted into the shRNA expression vector pGLVU6/Puro. The shRNA sequences (Supplementary Table 2) were designed to target the coding region of the mouse Khk-a (NM_001349066). The virus-containing supernatant was harvested from HEK293T cells, centrifuged at $800 \times g$, and filtered. The filtrates were incubated with Lentiviris Concentrator Solution containing PEG-8000 overnight at 4 °C. The filtrates were centrifuged at $1600 \times g$ for 60 min, and the viral pellet was re-suspended in PBS to be applied to MTV-TM/011 cells. The transfected cells were selected with 1 μg/mL of puromycin. To get stable cell lines for tumor grafting, several colonies of MTV-TM/011 were pooled. Nucleotide sequences of shRNAs are summarized in Supplementary Table 2.

**Breast-cancer xenografts**. All animal experiments were carried out with an approved protocol proposal from the Seoul National University Institutional Animal Care and Use Committee (approval No. SNU-170721-1-5; 190712-2-1; 190819-1). MTV-TM-011 cell lines harboring the CMV luciferase-IRES-GFP plasmid or the CMV luciferase-IRES-KHK-A plasmid were selected with G418; CMV luciferase-IRES-GFP plasmid with sh-Khk-a (or sh-control), or the CMV luciferase-IRES-KHK-C plasmid with sh-Khk-a were selected with G418 and puromycin; CMV luciferase-IRES-GFP plasmid with His-YWHAH_WT or His-YWHAH_S25A were selected with G418 and Zeocin. These cells were implanted into lower right mammary fat pads of female mice (8 weeks old, Balb/cSlc-nu/nu). Tab water or 15% fructose (or glucose)-containing water was provided ad libitum. To get live images of tumors, mice were anaesthetized with isoflurane and injected intraperitoneally with 100 μL of the VivoGlo luciferin (40 mg/mL) solution (Promega, Madison, WI). After 10 min, images were acquired with the Xenogen IVIS 100 and analyzed using the LivingImage 2.50.1 software (Xenogen, Alameda, CA). Two weeks after implantation, the primary tumor growth was monitored thrice a week. When average primary tumor volumes were reached 500–600 mm$^3$, the primary tumors were removed and tumor metastasis was monitored for 3 weeks. Major vital organs (lung, liver, heart, spleen, brain, kidney, hind leg, and GI tract) were collected to identify tumor metastasis.

**Cell invasion assay**. To assess cell invasion, the Transwell chambers partitioned by a Matrigel-coated membrane with 8 μm pore size was used. Cells were seeded onto upper chambers in serum-free medium at a density of $2 \times 10^4$ (H1299, A549, and MDA-MB-231), $3 \times 10^4$ (U87, U251, DU145, DLD-1, HCT116, 786O, RCC4, SiHa, and HeLa), $4 \times 10^4$ (MCF-7 and U2OS), $6 \times 10^4$ (MIApaca2, PanC1, and MTV-TM-011), or $7 \times 10^4$ (PC3 and HepG2) cells/well, an FBS-containing medium was placed in the lower chamber. After incubated for 18 h, cells were fixed with 4% paraformaldehyde and stained with Hematoxylin and Eosin. Cells on the upper side of the filters were removed with cotton-tipped swabs, and invaded cells on the downside were viewed and photographed under BX53-P polarizing microscope (Olympus, Tokyo, Japan). The invaded cells were counted using the ImageJ software (NIH, Bethesda, MD).

**Preparation of recombinant proteins**. Recombinant GST-KHK-A, GST-YWHAH, and free GST proteins were expressed in Escherichia coli BL21 cells, pulled down using glutathione-affinity beads (GE Healthcare; Chicago, IL) at 4 °C for 1 h, and eluted with 10 mM reduced glutathione (Sigma-Aldrich). FLAG-LRRC59 and His(6)-KHK-A proteins were expressed in HEK293T cells, pulled down using EZview Red anti-FLAG (Sigma-Aldrich) and nickel-NTA (Qiagen, Hilden, Germany) affinity beads at 4 °C for 4 h, and eluted with 500 ng/μL of FLAG peptide and 250 mM imidazole, respectively. The amounts and purities of proteins were checked by SDS-PAGE and Coomassie Brilliant Blue R-250 staining.

**In vitro binding assay**. The mixtures of purified His-KHK-A and FLAG-LRRC59 or GST-YWHAH were incubated in a binding buffer (25 mM HEPES/pH 7.5, 150 mM KCl, 12.5 mM MgCl$_2$, 0.5 mM DTT, 0.1% NP-40, and 10% glycerol) at 4 °C for 1 h with or without 1 mM fructose, and further incubated with nickel-NTA

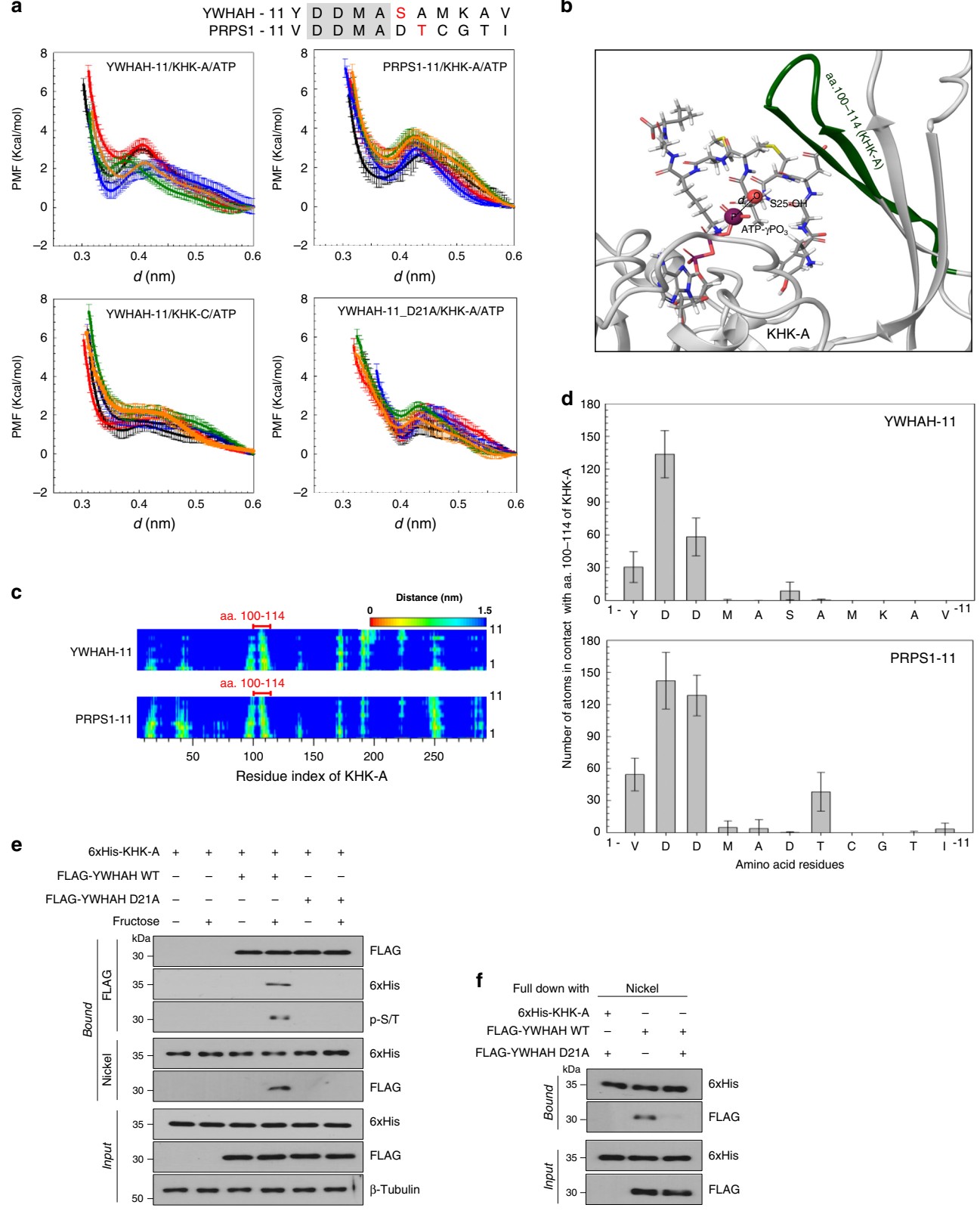

affinity beads at 4 °C for 1 h. After the beads were washed with the binding buffer, bound proteins were eluted in the denaturing SDS sample buffer and subjected to western blotting.

terminated by heating the samples in an SDS buffer. The protein phosphorylation was evaluated by western blotting.

**In vitro kinase assay**. Recombinant His-KHK-A (10 ng) or His-KHK-C (10 ng) and GST-YWHAH (500 ng) in 30 μL of the kinase buffer containing 50 mM Tris-HCl/pH 7.5, 100 mM KCl, 50 mM MgCl₂, 1 mM sodium orthovanadate, 1 mM DTT, 10% glycerol, and 2 mM ATP. The reactions were incubated at 37 °C for 1 h,

**Quantitative RT-PCR**. Total RNAs were extracted using TRIZOL (Invitrogen, Carlsbad, CA), and reverse-transcribed at 42 °C for 60 min in a reaction mixture containing M-MLV Reverse Transcriptase (Enzynomics, Daejeon, Korea), RNase inhibitor, dNTP, and random primers. Real-time PCR was performed in the qPCR Mastermix (Enzynomics), and fluorescence emitting from dye-DNA complex was

**Fig. 7 Molecular dynamics analysis to support the KHK-A phosphorylation of YWHAH. a** Potential of mean force (PMF) profiles as a function of the distance ($d$) between the oxygen of S25-OH and the phosphorus of ATP-$\gamma PO_3$ for YWHAH/KHK-A/ATP, PRPS1-11/KHK-A/ATP, YWHAH/KHK-C/ATP, and YWHAH-11_D21A/KHK-A/ATP. Amino acid sequences of YWHAH-11 and PRPS1-11 are shown above panel. The conserved amino acids are highlighted and the phosphorylated residues are red. For data reproducibility, five different PMF profiles for each case were obtained from fully independent umbrella simulations, which are marked with five different colors. The error bars were obtained from bootstrap analysis with 200 bootstraps. **b** 3D structure of YWHAH-11 bound to KHK-A. YWHAH-11 and ATP are shown by sticks with exceptions for the hydroxyl oxygen group of S25 and the $\gamma$-phosphorous of ATP represented by enlarged spheres between which distance is denoted as $d$ and KHK-A are represented by cartoons where aa. 100–114 are green. **c** The residue–residue contact map for ternary complex of YWHAH-11/KHK-A/ATP at $d \cong 3.5$ nm (top) and of PRPS1-11/KHK-A/ATP at $d \cong 3.6$ nm (bottom). The horizontal residues indicate aa. 3–298 of KHK-A and the right vertical residues indicate aa. 1–11 of YWHAH-11 or PRPS1-11. **d** The average number of atoms contacting the KHK-A 100–114 for YWHAH-11 residues at $d \cong 3.5$ nm (top) and PRPS1-11 residues at $d \cong 3.6$ nm (bottom). The contact is counted in case that the interatomic distance <5 nm. The averages were taken over $n = 1000$ configurations obtained from MD trajectories for 10 ns with 10 ps interval under the restraint of $d = 3.5$ nm (YWHAH-11) and $d \cong 3.6$ nm (PRPS1-11). Data are presented as means ± S.D. **e** MDA-MB-231 cells were co-transfected with His-KHK-A and FLAG-YWHAH WT (or D21A) plasmids. The protein interaction was cross-checked by co-precipitation using anti-FLAG and Nickel-NTA affinity gels. The Ser/Thr-phosphorylation of YWHAH was identified by immunoblotting the immunoprecipitated YWHAH with anti-phospho-S/T antibody. **f** In vitro binding assay. Purified FLAG-YWHAH WT (or D21A) and KHK-A were co-incubated for 1 h. KHK-A was pulled down using Nickel-NTA and the KHK-A-bound YWHAH was immunoblotted.

---

monitored in CFX Connect Real-Time Cycler (BIO-RAD, Hercules, CA). The mRNA values of targeted genes were normalized to GAPDH expression for each sample. All reactions were performed as triplicates. Primers used in real-time quantitative PCR were summarized in Supplementary Table 3.

**Chromatin immunoprecipitation**. Cells were fixed with 1% formaldehyde at 37 °C for 10 min, and washed with cold PBS containing protease inhibitors. After cells were lysed with 0.5% NP-40, the nuclear fraction was extracted with 1% SDS and sonicated to cut genomic DNAs into 300-500 bp fragments. Soluble chromatin complexes were precipitated with anti-FLAG, anti-SLUG, anti-SNAIL, or anti-YWHAH antibody overnight at 4 °C. Immune complexes were precipitated with protein A/G beads (Santa Cruz Biotechnology) at 4 °C for 2 h. The beads were sequentially washed with low salt, high salt, LiCl, and TE solutions. The bound chromatin complexes were eluted twice in a ChIP direct elution buffer for 15 min, and incubated overnight at 65 °C to reverse cross-linking. DNAs were extracted by phenol–chloroform and precipitated with ethanol. The precipitated DNAs were resolved in distilled water and analyzed by qPCR (95 °C/55 °C/72 °C, 20 s at each phase). The nucleotide sequences (5′ to 3′) of PCR primers are GCGCTGCTGAT TGGCTGTG and GGCTGGAGTCTGAACTGAC for the *CDH1* promoter. ChIP-qPCR results were represented as the percentages of the input signal (% input). All reactions were performed as triplicates.

**Fractionation of cytoplasmic and nuclear components**. Cells were centrifuged at $1000 \times g$ for 5 min, and homogenized in a lysis buffer containing 10 mM Tris/HCl (pH 7.4), 10 mM KCl, 1 mM EDTA, 1.5 mM MgCl₂, 0.2% NP-40, 0.5 mM DTT, 1 mM sodium orthovanadate, and 400 μM PMSF. The cell lysates were separated into pellets (for nuclear fraction) and supernatant (for cytosolic fraction) using centrifugation at $1000 \times g$ for 5 min. One packed volume of a nuclear extraction buffer (20 mM Tris/HCl, pH 7.4, 420 mM NaCl, 1 mM EDTA, 1.5 mM MgCl₂, 20% glycerol, 0.5 mM DTT, 1 mM sodium orthovanadate, and 400 μM PMSF) was added to the pellet, and vortexed intermittently at low speed on ice for 30 min. The nuclear and cytosolic fractions were spun at $20,000 \times g$ for 10 min and stored at −70 °C.

**Immunoblotting and immunoprecipitation**. Proteins were separated on SDS/PAGE and transferred to Immobilon-P membranes (Millipore, Bedford, MA). The membranes were blocked with 5% skim milk, incubated overnight at 4 °C with a primary antibody, incubated with a horseradish peroxidase (HRP)-conjugated secondary antibody for 1 h, and visualized using the ECL-plus kit (Amersham Biosciences, Piscataway, NJ). The uncropped blots were presented in Supplementary Fig. 14. To analyze protein interactions, cell lysates were incubated with anti-KHK, anti-YWHAH, or IgG overnight at 4 °C, and the immune complexes were pulled down with protein A/G beads. Otherwise, cell lysates were incubated with EZview Red anti-Myc, or anti-FLAG-affinity gel, or Nickel-NTA affinity beads at 4 °C for 4 h. The bound proteins were eluted in a denaturing SDS sample buffer and loaded on SDS-PAGE. All western blotting experiments were performed three or more times.

**Immunofluorescence analysis**. KHK-A and KHK-C cDNAs were fused to the C-terminus of GFP and observed using Carl Zeiss LSM510 META confocal microscope at 488 nm. To examine the subcellular localization of LRRC59 and YWHAH, cells were fixed with 3.7% formaldehyde for 10 min and permeabilized with 0.1% Triton X-100 for 30 min. Cells were incubated in PBS containing 0.05% Tween-20 and 3% BSA for 1 h, and further incubated overnight at 4 °C with a primary antibody. Cells were incubated with Alexa Fluor 594-conjugated secondary antibodies for 1 h. To stain F-actin, cells were incubated with Alexa Fluor 633 phalloidin (Invitrogen) for 30 min, stained with DAPI (Sigma-Aldrich) for 30 min,

mounted in Faramount aqueous mounting medium (Dako, Glostrup, Denmark), and examined under the confocal microscope.

**Immunohistochemistry**. Primary tumors and lungs from mice were fixed with 4% paraformaldehyde, paraffin-embedded, and cut into 4-μm section slices. The slides were incubated in a dry oven at 60 °C for 1 h, de-paraffinized, re-hydrated, and incubated in a citrate buffer (Dako) at 121 °C for 10 min to retrieve antigen. To block non-specific signals. the slides were incubated with 3% H₂O₂ for 15 min and with 2% horse serum in BSA solution for 1 h. The slides were incubated overnight at 4 °C with anti-KHK (1:100, Santa Cruz), anti-LRRC59 (1:100, Novus Biologicals), or anti-E cadherin (1:200, Thermo Fisher Scientific). The sections were biotinylated with a secondary antibody for 1 h. The immune complexes were visualized using the Vectastatin ABC kit (Vector Laboratories) and the DAB detection kit (Dako). Finally, the slides were counterstained with hematoxylin for 15 min, and photographed at four high-power fields in each slide.

**Mass analysis to identify protein interactome**. MCF-7 and MDA-MB-231 cells were transfected with an empty vector, His-KHK-A, or His-KHK-C, incubated with 5 mM fructose for 8 h. To identify protein interactions, cell lysates were incubated with nickel-NTA beads at 4 °C for 4 h. The bound proteins were eluted by 250 mM imidazole. Protein concentration of the elute was measured using the BCA kit (Thermo Fisher Scientific, Rockford, IL). After the Filter Aid Sample Preparation procedure was performed using a 30 K Amicon Ultra Centrifugal filter (Millipore, UK), peptides were separated on Nanoflow Easy-nLC1000 system (Proxeon Biosystems, Odense, Denmark), which was equipped with a packing column (100 Å, 1.8-μm particle, 75 μm × 50 cm). A gradient that ranged from 5 to 30% acetonitrile with 0.1% formic acid was run at a fixed flow rate of 300 nL/min for 90 min. Ionized peptides were analyzed on a quadrupole-orbitrap mass spectrometer (Q-Exactive, Thermo Fisher Scientific, San Jose, CA). MS1 spectra were measured at a resolution of 70,000 with an $m/z$ scan range of 300–1800. For MS2 spectra, peptides were fragmented by higher-energy collisional dissociation (HCD) with a normalized collision energy of 30 and a resolution of 17,500. Raw MS data files for the label-free quantification were processed using the Maxquant (ver.1.5.5.1), to match the spectra against the uniprotKB FASTA database (74,540 entries, version from June 2014). MS/MS peaks searches were performed with the following parameters: peptide length of at least six amino acids, fixed carbamidomethyl modification, variable methionine oxidation, and variable N-terminal acetylation. The tolerance was set to the 6 ppm, and 20 ppm for main search and first search, respectively. A false discovery rate of 1% was applied to all proteins and peptide searches.

**Mass analysis to identify post-translational modifications**. For LC-MS/MS analysis of intracellular YWHAH protein, MDA-MB-231 cells with KHK-A overexpression or knockdown were transfected with FLAG-YWHAH, and then were treated with 5 mM fructose for 24 h. FLAG-YWHAH was purified using FLAG-affinity beads and subjected to SDS-PAGE. For LC-MS/MS analysis of YWHAH phosphorylated in vitro, recombinant GST-YWHAH, which was reacted with GST-KHK-A as described in the in vitro kinase assay, was separated by SDS-PAGE. Proteins in gel slices were digested with trypsin and loaded to Easy n-LC (Thermo Fisher, San Jose, CA). Samples were separated on a C18 nanobore column (150 mm × 0.1 mm, 3 μm pore size; Agilent). The mobile phase A for LC separation was 0.1% formic acid, 3% acetonitrile in deionized water and the mobile phase B was 0.1% formic acid in acetonitrile. The chromatography gradient was designed for a linear increase from 5% B to 55% B in 40 min, 52% B to 75% B in 4 min, 95% B in 4 min, and 3% B in 6 min. The flow rate was maintained at 1500 nL/min. Mass spectra were analyzed using LTQ Orbitrap XL mass spectrometer (Thermo Fisher) equipped with a nano-electrospray source. Mass spectra were acquired using data-dependent acquisition with a full mass scan (350–1200 $m/z$) followed by 10 MS/MS

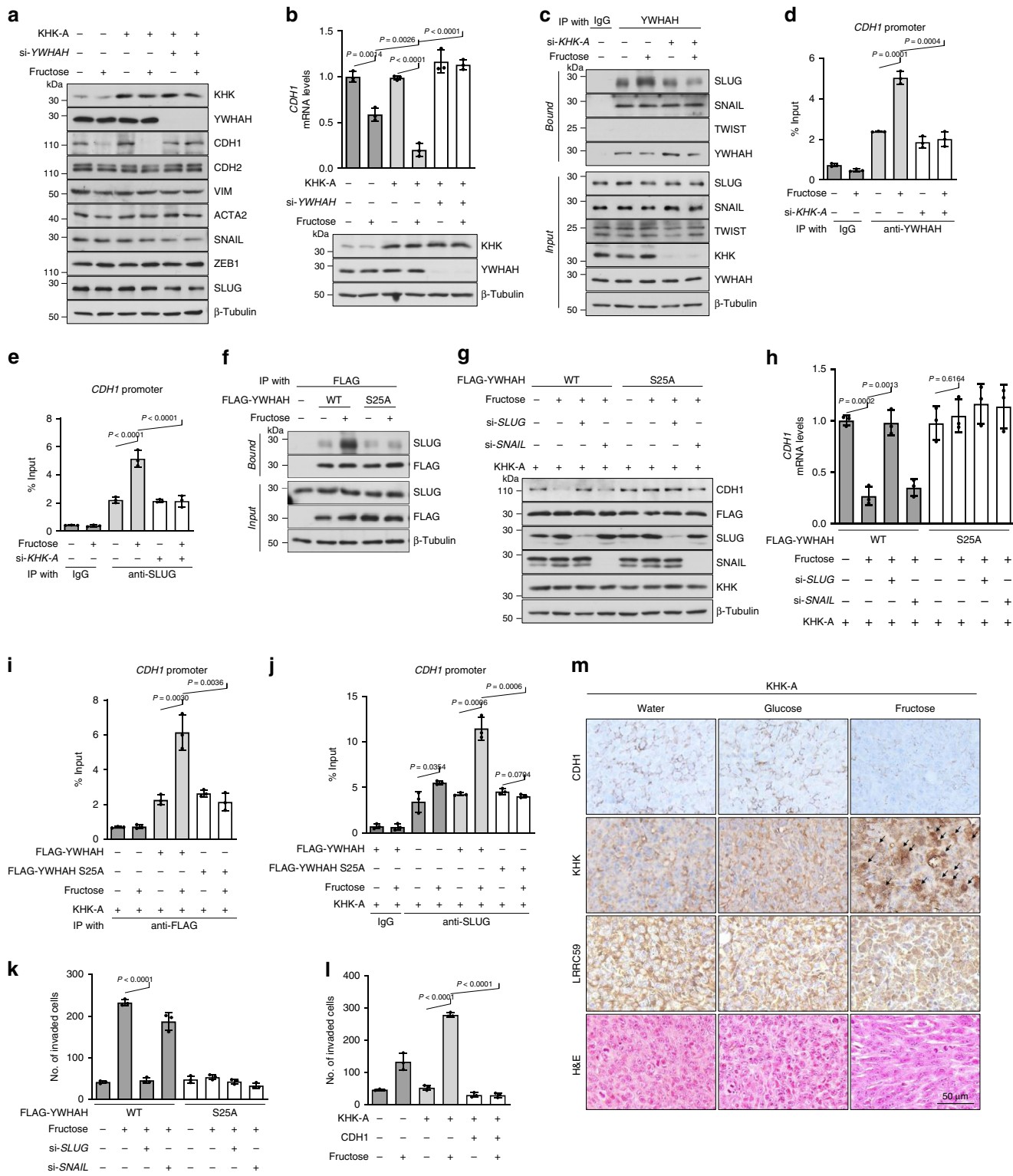

scans. For MS1 full scans, the orbitrap resolution was 30,000 and the AGC was 2 × 10⁵. For MS/MS in the LTQ, the AGC was 1 × 10⁴. The phosphorylation was identified by the additional mass of 78 Dalton on digested peptides. To identify modified amino acid, proteome discoverer (V1.2.0.208 with SEQUEST algorithm) was used. MS/MS spectra and detected fragment ions are represented in Supplementary Figs. 15 and 16.

**Mass analyses of fructose and fructose-1-phosphate**. MCF-7 and MDA-MB-231 cells were incubated with or without 5 mM fructose for 4 h. Cells were gently homogenized in a hypotonic solution with 0.6% NP-40 on ice, and spun down at

1000 × g, and the pellet was saved as the nuclear fraction. The supernatant was centrifuged at 10,000 × g and collected as the cytoplasmic fraction. The cytoplasmic and nuclear fractions were vortexed with an ice-cold extraction solvent (acetonitril: methanol:water = 4:4:2, v/v/v) for 1 min and snap-frozen in liquid nitrogen for 1 min, followed by centrifugation at 10,000 × g for 15 min. The samples were evaporated using SpeedVac vacuum centrifugation. Dried samples were derivatized with 50 µL of methoxyamine hydrochloride in pyridine (20 mg/mL) at 30 °C for 90 min, followed by trimethylsilylation with 70 µl of N-methyl-N-(trimethylsilyl) trifluoroacetamide at 60 °C for 30 min. The derivatized samples were applied to Leco LECO Pegasus® BT time-of-flight (TOF) MS (Leco Corporation, Edinburgh, UK) coupled with an Agilent 6890 GC (Agilent Technologies) equipped with a

**Fig. 8 S25-phosphorylated YWHAH promotes the SLUG recruitment to the CDH1 promoter. a** Transfected cells were incubated with fructose, and subjected to EMT markers analysis. **b** The mRNA levels of *CDH1* were analyzed by RT-qPCR. **c** Protein binding analysis between YWHAH and endogenous YWHAH SLUG, SNAIL, or TWIST. **d** The fructose-induced YWHAH recruitment to the CDH1 promoter depends on KHK-A. After MDA-MB-231 cells were transfected with si-*KHK-A*, incubated with fructose, then subjected to ChIP-qPCR using anti-YWHAH antibody. **e** The fructose-induced SLUG recruitment to the *CDH1* promoter depends on KHK-A. **f** The S25 phosphorylation of YWHAH is essential for YWHAH binding to SLUG. **g** The S25 phosphorylation of YWHAH is essential for the fructose- and SLUG-dependent suppression of E cadherin. **h** The mRNA levels of *CDH1* were analyzed by RT-qPCR. **i** FLAG-YWHAH binding to the CDH1 promoter was analyzed by ChIP-qPCR. **j** The S25 phosphorylation of YWHAH is essential for the SLUG binding to the *CDH1* promoter. Transfected MDA-MB-231 cells were incubated with fructose. The chromatins containing endogenous SLUG were immunoprecipitated with anti-SLUG antibody and the SLUG-bound CDH1 promoter was quantified by real-time PCR. Results were presented as the percentages of input data. **k** The S25 phosphorylation of YWHAH is essential for the fructose-induced, SLUG-dependent cell invasion. **l** The fructose-induced, KHK-A-dependent cell invasion was attenuated by CDH1 restoration. **m** Immunohistochemical analyses of the breast-cancer tissues presented in Fig. 2. The tumor sections were immunostained with the antibodies against CDH1, KHK, and LRRC59, and stained with Hematoxylin and Eosin. In (**a–l**), experiments performed using MDA-MB-231 cell line. In (**a**), (**b**), (**g**), (**h**), (**k**), (**l**), transfected cells were incubated with 5 mM fructose for 48 h, and In (**c–f**), (**i**), (**j**), cells were incubated with 5 mM fructose for 24 h. In (**b**), (**d**), (**e**), (**h–l**), data represented as the means ± S.D. from n = 3 independent experiments. Significance was calculated by unpaired, two-sided Student's *t*-test.

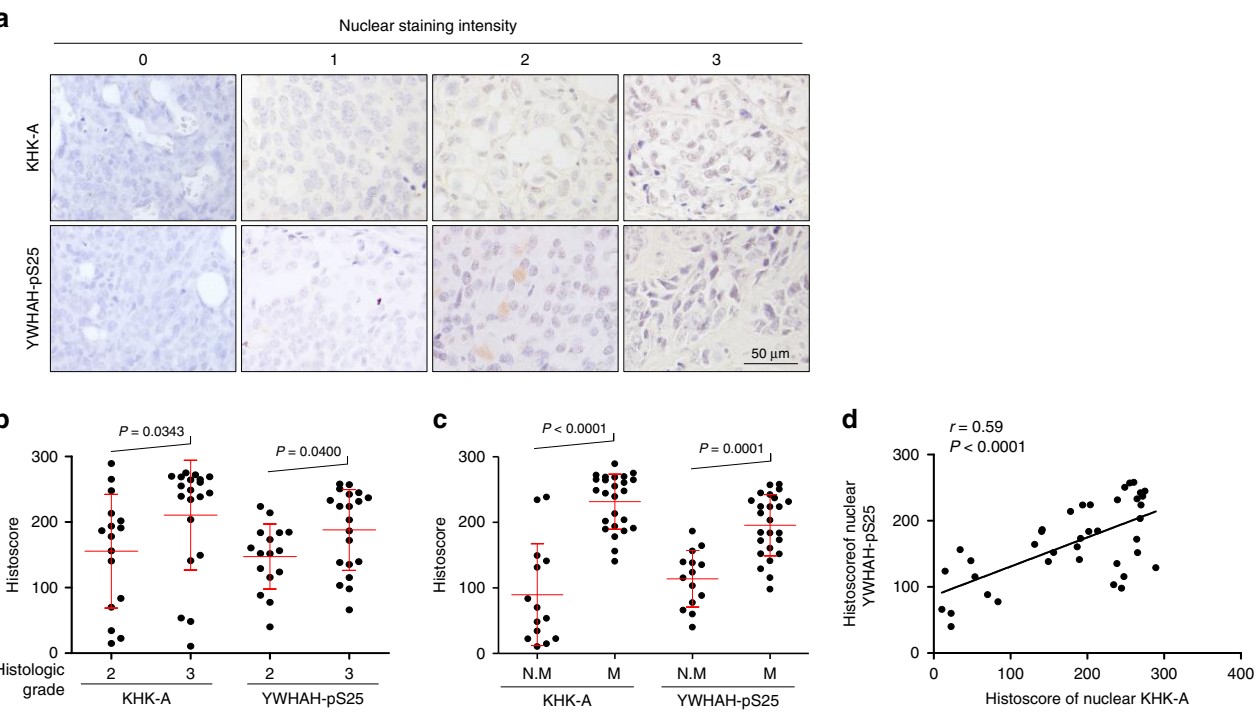

**Fig. 9 The YWHAH phosphorylation at S25 is associated with breast-cancer metastasis. a** Representative microscopy photographs of immunohistochemical staining in breast-cancer tissue arrays. **b** The nuclear levels of KHK-A and YWHAH-pS25 were compared between histologic grade 2 and 3 groups (the means ± S.D. from 36 different breast-cancer tissues). **c** Nuclear KHH-A and YWHAH-pS25 levels were compared between the non-metastatic group (N.M) and the metastatic groups (M) including lymphatic and distant metastases (the means ± S.D. from 36 different breast-cancer tissues). In (**b**), (**c**), significance was calculated by two-sided Mann–Whitney *U* test. **d** Spearman's correlation analysis between nuclear KHK-A and YWHAH-pS25 levels. 'r' is the correlation coefficient.

30-m-long 0.25 mm Rtx-5Sil MS column. The chromatographic condition was a constant flow of 1 mL/min and the temperature was increased from 50 to 330 °C at the rate of 20 °C/min. The electron ionization and ion source temperature were set −70 eV and 250 °C, respectively. To measure fructose-1-phosphate, cells were harvested in 500 μL of ice-cold ethanol, vigorously vortexed with 400 μL of chloroform, mixed with 150 μL of water, and centrifuged at 10,000 × g for 15 min. The upper phase containing polar metabolites was subjected to derivatization and applied to GC-MS analysis, as described above. Fructose-1-phosphate was quantified using the standard chemical (MF03840) provided by Carbosynth LTD (Berkshire, UK).

**Kinetic parameters for the KHK-A-YWHAH reaction.** Ten ng of recombinant GST-KHK-A and various amounts of GST-YWHAH were incubated with various concentrations of fructose in 51 μL of a kinase buffer containing 1 mM sodium orthovanadate, 2 mM ATP, and 10 μCi of [γ-$^{32}$P]-ATP. The reactions were incubated at 37 °C for 10 min, and terminated by adding the 10% ice-cold Trichloroacetic acid. Precipitated proteins were spun down at 15,000 × g, and washed three times in ice-cold acetone. The radioactivity in the pellet was measured with a

scintillation counter. The $K_m$ of KHK-A for YWHAH was calculated from a plot of 1/c.p.m. versus 1/[YWHAH] according to the Lineweaver–Burk equation. The $K_i$ of fructose for the KHK-A-mediated YWHAH phosphorylation was fitted with the competitive inhibition model from the data using the GraphPad Prism 5.0 software. Data represented the means from three independent experiments. To measure the IC$_{50}$ of fructose against the KHK-A-YWHAH reaction, 10 ng of GST-KHK-A and 1 μg of GST-YWHAH were incubated with various concentrations of fructose in 51 μL of the reaction buffer at 37 °C for 10 min. The IC$_{50}$ was calculated from a plot of KHK-A enzyme activity versus log[fructose] using the GraphPad Prism 5.0 software. Data represented the means from three independent experiments.

**Determination of $K_d$ of KHK-A and KHK-C for fructose.** Recombinant His-KHK-A or His-KHK-C (10 ng) and various concentrations of D-[$^{14}$C(U)]-fructose (ARC 0116A, American Radiolabeled Chemicals, St. Louis, MO) were mixed in 30 μL of 25 mM HEPES or 50 mM phosphate buffer solution (pH 7.5) containing 85 mM KCl, 17 mM NaCl, 1 mM MgCl$_2$, and 0.2 mM CaCl$_2$. The mixtures were incubated at 37 °C for 10 min and the reactions were terminated by adding the 10%

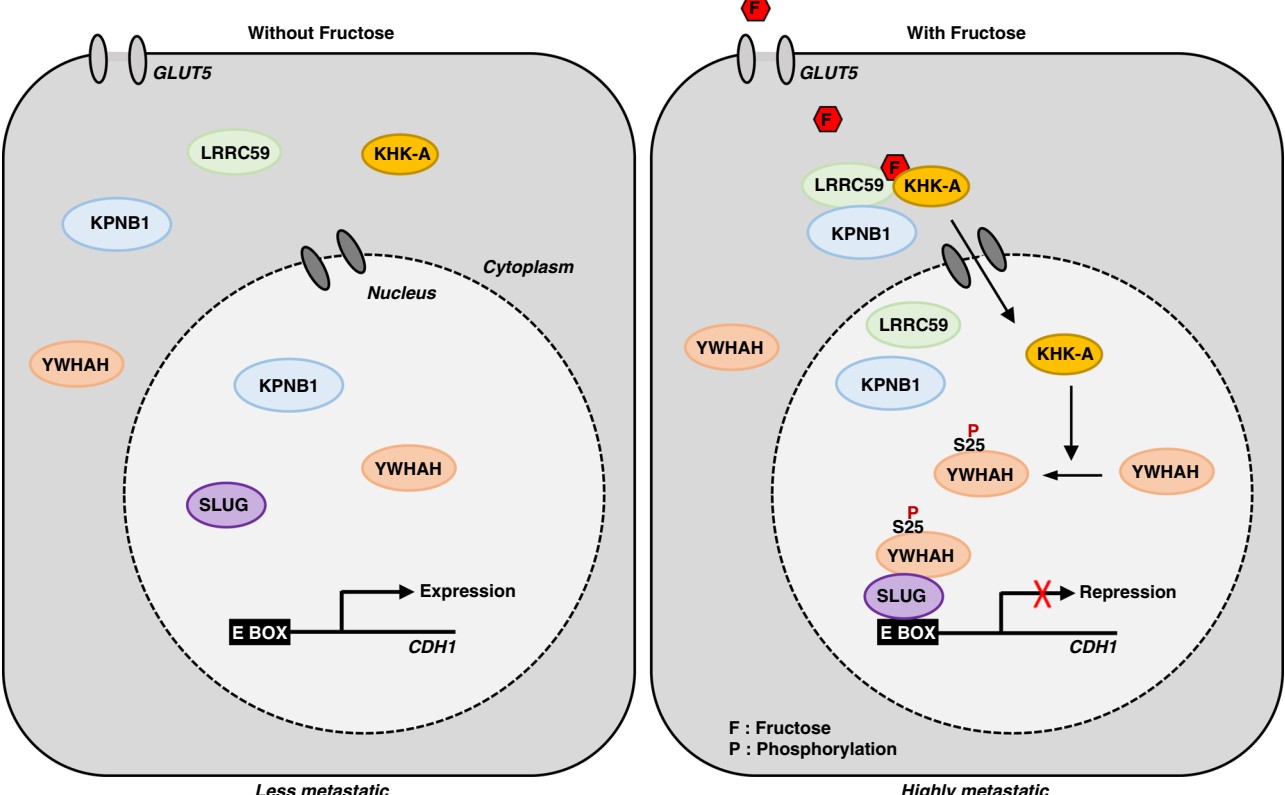

**Fig. 10 The proposed mechanism underlying fructose-induced cancer metastasis.** Nuclear translocalization of Ketohexokinase-A triggered by fructose stimulation, which is accompanied by the interaction to nuclear importer LRRC59 and KPNB1. In the nucleus, KHK-A phosphorylates YWHAH at Ser25, and in turn, YWHAH recruits SLUG to the CDH1 promoter, in turn negatively regulates of expression of E Cadherin. The KHK-A signaling triggered by fructose promotes cell migration, consequently, cancer cells acquire metastatic power.

ice-cold trichloroacetic acid. Precipitated proteins were spun down at $15,000 \times g$, and washed three times in ice-cold acetone. The radioactivity of protein-bound fructose in the pellet was measured with a scintillation counter, and the $K_d$ for fructose was fitted using the GraphPad Prism 5.0 software.

**Human breast-cancer tissue microarray.** Human breast-cancer tissue microarrays, which contained 40 different breast-cancer specimens, were purchased from SuperBioChips Lab (Seoul, South Korea). Clinical records, including sex, age, TNM stage, histologic grade, survival, and cause of death, were also provided by the company. Histologic grading was defined according to the Nottingham Modification of Bloom-Richard System. Clinical information is summarized in Supplementary Table 4. To score protein expression in immunostained specimens, two examiners observed four high-power fields on each slide. Protein expression was analyzed using histoscore, which reflects both the intensity of nuclear staining (graded as 0, non-staining; 1, weak; 2, moderate; 3, strong) and the percentage of positive cells. The range of possible scores was from 0 to 300. Clinical information is summarized in Supplementary Table 4.

**ROS measurement.** MCF-7 and MDA-MB-231 cells, which had been transfected with Myc-KHK-A, Myc-KHK-C plasmid, or siRNA, were incubated 5 mM fructose with or without 400 nM of pyrimidinopyrimidine, $20 \mu M$ of $H_2O_2$ was used as positive control for ROS production. After incubation with fructose, DCF-DA ($20 \mu M$) was added to cells. After being incubated in the dark for 20 min, cells were washed twice in PBS, and applied to a TECAN infinite M200 pro (Grödig, Austria). DCF fluorescence was excited at 488 nm and detected at 524 nm.

**Molecular dynamics simulation.** All-atom atomistic molecular dynamics (MD) models were used to simulate ternary complex system of YWHAH-11/ATP/KHK-A. The KHK-A/ATP and YWHAH-11 as a receptor–ligand pair were prepared by modifying KHK-A/fructose/adenyl-50-yl imidodiphosphate and YWHAH obtained from PDB ID: 2hw1 and 2c63, respectively[29,35]. Molecular docking was carried out using Hex package (version 8.0.0)[36] which generates the ensemble of ligand (YWHAH-11)-receptor(KHK-A/ATP) conformations by updating randomly the coordinates of the ligand and receptor molecules via spherical polar Fourier algorithm[37]. During the docking process, the ligand and receptor molecules were kept rigid. Among the produced ligand-receptor structures, the most favorable structure with the highest energy score was selected for the initial structure.

The selected structure was then solvated by TIP3P water molecules with 100 mM NaCl in the simulation box with a periodic boundary condition. The solvated system was then equilibrated for 10 ns at 310 K and 1 bar by performing NPT-ensemble MD simulations using GROMACS package[38] (version 5.1.2) with CHARMM36 force field[39]. The modified Berendsen thermostat[40] and Parrinello–Rahman barostat[41] were used for maintaining temperature and pressure. Following equilibration, starting from the end of the trajectory obtained from the previous equilibrium simulation, a steered MD simulation was performed by pulling the oxygen of S25-OH of YWHAH-11 toward the phosphorus of ATP-$\gamma PO_3$ over 1 ns using spring constant of 1000 kJ mol$^{-1}$ nm$^{-2}$ and a pull rate of 0.001 nm ps$^{-1}$ to get a trajectory of S25-OH approaching to ATP-$\gamma PO_3$ where ATP was fixed as an immobile reference by position restraint algorithm. Using this trajectory from pulling simulation, configurations having different $d$ with interval 0.1 nm were prepared as starting configurations for umbrella sampling where 10 ns of NPT-ensemble MD were subsequently performed. The PMF profile was finally obtained by the weighted histogram analysis method (WHAM)[42]. We performed five fully independent sets of umbrella simulations for each of the profiles to check the reproducibility. Furthermore, the uncertainties of the profiles were also estimated by resampling technique using Bayesian bootstrap analysis with 200 bootstraps. The whole simulation procedure described above was conducted also for YWHAH-11/ATP/KHK-C, PRPS1-11/ATP/KHK-A, and YWHAH-11_D21A/ATP/KHK-A to compute their PMFs.

**KHK isoform analysis from TCGA dataset.** KHK-A and KHK-C expressions in human cancers were analyzed by using data from The Cancer Genome Atlas (TCGA). Primary data (rnaseq V2 level 3 data for transcript isoforms normalized by RSEM) downloaded from the Broad Institute FireBrowse portal[43]. The UCSC isoform identifiers against the *KHK* gene were discriminated in the UCSC genome browser by the position of exon 3 (uc002ril.3 for KHK-A; uc002rim.3 for KHK-C).

**Statistical analysis and reproducibility.** All data were analyzed using the Microsoft Excel 2013 or the GraphPad Prism 8.0 software. Results were presented as the means and standard deviation (S.D.). The unpaired, two-sided Student's *t*-test was used to analyze the results of cell number, RNA level, protein level, and mass-based interactome data. A two-sided Mann–Whitney *U* test was used to analyze the results of histoscore, ROI flux, tumor volume, mouse body weight, and Spearman correlation analysis used to analyze the correlation coefficient between

KHK-A and YWHAH-pS25 expression in breast-cancer tissue arrays. The statistical significance was considered when $P < 0.05$. All immunoprecipitation (Figs. 3c, d; 4e, f, h; 5b, e; 7e, f; 8c, f; Supplementary Figs. 7b, c; 8g, h, j; 9c; 10a, b, c; 12e, f, k), immunoblotting (Figs. 3c, d; 4a, b, d, e, f, g, h; 5a, b, c, e; 7e, f; 8a, c, f, g; Supplementary Figs. 3b, e; 7a; 8a, b, c, d, e, g, h, i, j, k; 9b, c, d, i; 10a, b, c, d, e, f; 12a, b, e, f, j, k, l), immunohistochemistry and H&E staining (Figs. 2f; 6f; 8m; 9a; Supplementary Figs. 6g; 13a, b, c), immunofluorescence (Figs. 1e, g; 4c, i; 5f; Supplementary Figs. 8e; 9a), RT-qPCR, ChIP-assay, in vitro binding assay (Figs. 4d, 5a), in vitro kinase assay (Fig. 5c; Supplementary Fig. 9d), invasion assay (Supplementary Figs. 1; 3a, b, e; 4a, b, c; 7a; 8l; 9j, k; 12p, q, r, s), and proteomics analyses were independently repeated at least three times. The results for significance tests are included in each panel.

**Reporting summary**. Further information on research design is available in the Nature Research Reporting Summary linked to this article.

## Data availability

The mass spectrometry proteomics data have been deposited to the ProteomeXchange Consortium via the PRIDE partner repository with the dataset identifier PXD021035. The proteomics data referenced during the study are available in a public repository from the PRIDE website. All the other data supporting the findings of this study are available within the article and its supplementary information files and from the corresponding author upon reasonable request. A reporting summary for this article is available as a Supplementary Information file.

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

## Acknowledgements

This study was supported by the National Research Foundation of Korea (2019R1A2B5B03069677 and 2020R1A4A2002903).

## Author contributions

J.W.P. supervised the study. J.Kim and J.Kang designed and performed most of the experiments with assistance from Y.L.K. J.W. and Y.K. performed LC-MS/MS and

analyzed proteomics data. J.H. analyzed molecular dynamics. J.Kim, J.Kang, and J.W.P. wrote the manuscript. All authors commented on the manuscript.

## Competing interests

The authors declare no competing interests.
