## [Peer Review File · Nature Communications]

Editorial Note: STR Profile Results on p15 in this Peer Review File are reproduced with permission from Korean Cell Line Bank.

Reviewers' comments:

Reviewer #1, Expertise : cancer metabolism, metastasis(Remarks to the Author) :

The authors showed that fructose triggers the binding of KHK-A to LRRC59 and KPNB1 and subsequent translocation of KHK-A to the nucleus, where KHK-A phosphorylates YWHAH. This phosphorylation is critical for YWHAH to interact SLUG and the binding of SLUG to the CDH1 promoter, leading to downregulation of E-cadherin and promoted tumor cell invasion and metastasis.

The finding that nuclear KHK-A has protein kinase activity and regulates E-cadherin expression through phosphorylation of YWHAH is interesting and novel. However, how fructose promotes nuclear accumulation of KHK-A but not KHK-C is not investigated. Given that KHK-A has limited binding affinity to fructose, how fructose triggers a signaling to affect KHK-A cellular redistribution is of significance to understand the role of fructose metabolism in cancer progression.

Specific points:

1. Is fructose still metabolized in KHK-A- but not KHK-C-expressed cancer cells? A fructose flux experiment is required for further elucidating the situation of fructose metabolism in the cancer cells.
2. Fig.1 and Supplementary Fig. 1a. These experiments are not rationally designed. The conclusion that fructose but not glucose promotes cell invasion is misleading. It is well known that glucose is critical for tumor cell migration, invasion, and proliferation. The authors should add an equal amount of fructose and glucose into glucose- and fructose-free medium when doing these experiments.
3. Fig. 1d and Supplementary Fig. 1c. Immunoblot showing successful depletion of these proteins should be performed. The cells with KHK-A depletion should have a reconstituted expression of KHK-C so that the effect of KHK-C on rescue of KHK-A depletion can be examined.
4. Fig. 1g,h and Supplementary Fig. 1f show that KHK-specific kinase inhibitor pyrimidinopyrimidine (Pypy) inhibited cell invasion. Is the observed effect due to inhibition of fructose metabolism? This possibility should be examined.
5. "However, the knockdown of either ALDOB or ALOX12 failed to attenuate cell invasion upon fructose stimulation (Supplementary Fig. 1g,h)". These results are inconsistent with a previous publication. A discussion is needed.
6. Fig. 2. Animal studies should include injection of cells with KHK-A depletion and KHK-A depletion with reconstituted expression of KHK-C.
7. Fig. 3C, 3D. IP with specific antibodies against KHK-A or KHK-C should be used.
8. Fig. 4A, 4B. Immunoblots with specific antibodies against KHK-A or KHK-C should be used.
9. Does fructose treatment affect the binding of KHK-A to PRPS1 and PRPS1 phosphorylation by KHK-A since some of KHK-A translocates to the nucleus?
10. If leucine 83 of KHK-A is involved in binding to both PRPS1 and KPNB1/LRRC59, how does fructose regulate the binding of to KHK-A to PRPS1 versus to KPNB1/LRRC59?
11. Fig. 5. KHK-C should be included as a control in the protein kinase assay.
12. Fig. 5C shows that fructose inhibits KHK-A-dependent phosphorylation of YWHAH. The IC₅₀ and K_i of fructose on inhibition of KHK-A-mediated phosphorylation of YWHAH should be measured. What is the nuclear concentration of fructose when the cells treated with fructose? Will KHK-A be inhibited by nuclear fructose under this concentration?
13. 14-3-3 η (YWHAH) is also a cytosolic protein. Why is KHK-A unable to phosphorylate YWHAH in cytosol?
14. Fig. 5G. Expression of Flag-tagged YWHAH proteins only limitedly restored the expression of YWHAH. Why did the cells with Flag-tagged WT YWHAH expression have much more cell invasion than their parental cells?
15. Fig. 6. Does D220 in PRPS1 abolish the binding of PRPS1 to KHK-A and PRPS1 phosphorylation by KHK-A?
16. Fig. 7. Does expression of YWHAH S25A blocked tumor metastasis in mouse?
17. A recent study (Xu et al., Science Advances, 2019) showed that KHK-A phosphorylates p62 promote tumor cell survival? Does adding fructose in medium alter p62 phosphorylation by KHK-A?

Reviewer #2, Expertise: Breast cancer, EMT and metastasis (Remarks to the Author):

In this manuscript, the authors described a new mechanism by which KHK-A mediates the fructose-induced epithelial-mesenchymal transition and invasion of breast cancer cells. Mechanistically, KHK-A interacts with LRRC59 and KPNB1 in fructose dependent manner, and thereby is translocated into nucleus where it phosphorylates YWHAH. Phosphorylated YWHAH recruits Slug to the promoter region and represses the transcriptional activity of CDH1 gene. The experiments are well-designed with rigorous controls, and the data well support their hypothesis. However, the clinical evidence is lacking and the physiological significance of KHK-A signaling is not fully addressed, which are needed to better justify the translational potential of their findings. Major comments:

1) The relationship between fructose consumption and metastatic progression of breast cancer has not been observed or reported, though such connection has been studied on mouse models. The lacking of clinical evidence of the fructose/KHK-A/YWHAH axis in promoting breast cancer metastasis is the major weakness of this manuscript. To justify the clinical importance of their findings, it will be better to investigate the relationship among KHK-A expression, YWHAH phosphorylation and metastasis progression in patient samples.

2) The major findings of fructose/KHK-A induced EMT, such as down-regulation of both protein and mRNA level of CDH1, and ChIP assay of SLUG binding on CDH1 promoter, were test only on MDA-MB-231 cells. MDA-MB-231 cells are well known to be an invasive breast cancer cell line with extremely mesenchymal phenotype. It is not common to further induce EMT in such cells. In addition, EMT is known to lead to the loss of epithelial markers accompanied with gain of mesenchymal markers. However, as shown in Figure 1f, neither fructose treatment alone or with over-expressed KHK-A could increase the expression levels of mesenchymal markers in MDA-MB-231 cells. Therefore, another epithelial cell line, such as MCF7, is required to validate the effects of fructose/KHK-A on EMT.

3) The current design of in vivo study is less clinically relevant. The metastasis promoting effect of fructose drink was only observed in the group with ectopic expression of KHK-A but not in the control group with the physiological level of KHK-A. It would be better to include another design showing that loss of function of KHK-A could diminish the metastasis in vivo.

Minor comments:

1) Figure 1b, the absolute quantitation qRT-PCR of KHK isoforms should be used to compare KHK-A and KHK-C expression levels in cells. The relative quantitation qRT-PCR may be misleading due to the difference in the amplification efficiency of KHK-A and KHK-C primers.

2) MDA-MB-231 cells are typically spindle shaped morphology in monolayer culture (<https://www.atcc.org/products/all/HTB-26.aspx#characteristics>), and have high expression level of vimentin but very weak or even non-detectable protein levels of both E-Cadherin and N-Cadherin as determined by western blotting (PMID: 10545506). The current immunofluorescence and immunoblotting data are not consistent among different panels or with the above mentioned impression on MDA-MB-231 cells. A cell authentication report may be required to prove the identity of MDA-MB-231 used in this study.

3) Several references are not properly cited. For example, neither literature 5 nor 6 discussed the association of fructose consumption and risk of multiple types of cancer except the endometrial cancer. Literature 18 does not report that breast cancer cells can metastasize in response to fructose.

4) Both YWHAH and KHK-A proteins are also present in the cytosolic compartment, as suggested by Figure 3D. Thus, the figure 8 is not accurate and kind of misleading. The mechanism underlying the observation that KHK-A does not interact with cytosolic YWHAH upon fructose treatment may illuminate new strategies to block this signaling axis.

Reviewer #3, Expertise: molecular modelling and dynamics (Remarks to the Author):

The paper investigates the molecular mechanisms at the basis of fructose intake and breast cancer metastasis, supporting clinical evidence for fructose-induced cancer aggravation. The authors present evidences that KHK-A signaling pathway is involved in fructose-induced cell invasion. In particular, they show molecular experiments supporting the evidence that ketohexokinase-A, rather than ketohexokinase-C, is necessary and sufficient for fructose-induced cell invasion.

While I find the paper interesting and well organised, with a nice combination of computational and experimental results, I have some major reservations that, I think, are in the way for a clear conclusion on the reported data.

The evidence that the main target is KHK-A and not other isoforms is based on observed overexpression of KHK-A in a number of relevant cell lines, but should be better supported.

A recent paper (not cited by the authors) Clin Invest. 2018;128(6):2226-2238.

<https://doi.org/10.1172/JCI94427>

present convincing evidence that protection from High Fructose Intolerance (HFI) is resulting from inhibition of the KHK-C isoform, and not the A. So the mechanistic interplay of these two Kinases seems to have a prominent role with respect to fructose intake. Once silenced KHK-A what happens to the expression on KHK-C in vivo? We do not know if these isoforms would replace the KHK-A activity in such condition and therefore we cannot use the argument of specifically inhibiting KHK-A. Moreover how specific is the named inhibitor is not clarified and not texted here with respect to the fructose signalling pathways. The authors should show the results also with less specific inhibitors, and if available, inhibitors targeting KHK-C vs KHK-A.

I have some major reservations also on the computational data here reported.

The docking and pulling simulations are very poorly described. The intent of the pulling simulation was to describe the Potential of mean force (PMF) associated to the process of S25-OH approaching ATP-gammaPO3 and therefore support the hypothesis that YWHAH is phosphorylating KHK-A (Fig 6 panel a). First of all the description of the residue contacts playing a role in this process is difficult to follow and the comparison with PRPS11 is not clearly discussed and documented.

Moreover the PMF is reported as a single profile for each experiment and there is no evidence of the reproducibility or of the error associated to these. Multiple replica of the pulling process are usually needed to validate the observations.

Additionally, resampling with the WHAM method, for example, would at least give some confidence to the observed overlap or differences in the profiles.

Reviewer #1

1. Is fructose still metabolized in KHK-A- but not KHK-C-expressed cancer cells? A fructose flux experiment is required for further elucidating the situation of fructose metabolism in the cancer cells.

Answer: Thanks for such a critical point. We examined the possibility that KHK-A drives the fructose flux to generate fructose-1-phosphate and ROS. Consequently, fructose supplementation increased fructose-1-phosphate and ROS levels in MDA-MB-231 and MCF-7 cells ectopically expressing KHK-C, but not in the cells expressing KHK-A. These results indicate that KHK-A does not participate in fructose metabolism.

Revision: Supplementary Fig 3f and 3g. Lines 20-24 on p5; Lines 1-6 on p28; Lines 12-18 on p29.

2. Fig.1 and Supplementary Fig. 1a. These experiments are not rationally designed. The conclusion that fructose but not glucose promotes cell invasion is misleading. It is well known that glucose is critical for tumor cell migration, invasion, and proliferation. The authors should add an equal amount of fructose and glucose into glucose- and fructose-free medium when doing these experiments.

Answer: As suggested, we added an equal amount (finally 5 mM) of fructose or glucose to glucose/fructose-free medium and performed the invasion assay. We found that breast cancer cells get more invasive under fructose stimulation but not under glucose stimulation. Given previous reports, hyperglycemia (>25 mM glucose) seems to facilitate cancer cell migration. As this work is not related with diabetes, we here have no reason for checking the hyperglycemia effect on breast cancer metastasis.

Revision: Supplementary Fig. 3a. Lines 2-5 on p5.

3. Fig. 1d and Supplementary Fig. 1c. Immunoblot showing successful depletion of these proteins should be performed. The cells with KHK-A depletion should have a reconstituted expression of KHK-C so that the effect of KHK-C on rescue of KHK-A depletion can be examined.

Answer: We added the western blots to verify knock-down and expression efficiencies of target proteins.

Revision: Supplementary Fig. 3b (the left panel). Line 9 on p5.

4. Fig. 1g,h and Supplementary Fig. 1f show that KHK-specific kinase inhibitor pyrimidinopyrimidine (Pypy) inhibited cell invasion. Is the observed effect due to inhibition of fructose metabolism? This possibility should be examined.

Answer: To examine the fructose flux, we measured fructose-1-phosphate and ROS levels. The fructose-induced enhancements of fructose-1-phosphate and ROS were observed in breast cancer cells expressing KHK-C, and those were inhibited by Pypy. However, the KHK-A-expressing cells did not produce fructose-1-phosphate and ROS in response to fructose regardless of Pypy treatment. Thus, we concluded that the inhibitory effect of Pypy against invasion of KHK-A-expressing cells is not attributed to the inhibition of fructose metabolism.

Revision: Supplementary Fig. 3f and 3g. Lines 20-24 on p5; Lines 1-6 on p28; Lines 12-18 on p29.

5. “However, the knockdown of either ALDOB or ALOX12 failed to attenuate cell invasion upon fructose stimulation (Supplementary Fig. 1g,h)”. These results are inconsistent with a previous publication. A discussion is needed.

Answer: We tried to test possibility of the involvement of pre-mentioned proteins with fructose-induced breast cancer metastasis (Supplementary Fig 4a and b). The inconsistent results with previous reports may be due to differences in cell lines and experimental conditions.

Revision: Lines 3-5 on p6

6. Fig. 2. Animal studies should include injection of cells with KHK-A depletion and KHK-A depletion with reconstituted expression of KHK-C.

Answer: We performed in vivo studies with sh-*Khk-a* deficient cancer cells with/ or without expressing KHK-C.

Revision: Supplementary Fig. 6. Lines 15-16, 18, and 22-23 on p6. Lines 1-2 and 4-9 on p7; Lines 1-10 on p20; Lines 15-20 on p20

7. Fig. 3C, 3D. IP with specific antibodies against KHK-A or KHK-C should be used.

Answer: We newly performed immunoprecipitation experiment with KHK-A or KHK-C specific antibodies.

Revision: Fig 3C & 3D. Lines 20-25 on p33.

8. Fig. 4A, 4B. Immunoblots with specific antibodies against KHK-A or KHK-C should be used.

Answer: We performed Western blotting with KHK-A or KHK-C specific antibodies.

Revision: Fig 4A & 4B, Supplementary Fig 8C & 8D. Lines 6-8 p8; Lines 3-5 on p34.

9. Does fructose treatment affect the binding of KHK-A to PRPS1 and PRPS1 phosphorylation by KHK-A since some of KHK-A translocates to the nucleus?

Answer: We checked these points in three different cell lines, as shown in the bottom. The phosphorylation of PRPS1 appears to be slightly reduced in breast cancer cell lines treated with fructose, but not in Hep3B hepatoma cells. We also checked the protein interaction. Fructose does not affect the interaction between PRPS1 and KHK in all cell lines tested. Therefore, we thought that fructose may functionally inhibit the KHK-A phosphoreaction of PRPS1 cell context-dependently. Even though we got these data, we do not want to include them in the present study focusing on YWHAH because they are not essential in this work.

Hep3B, MCF-7 and MDA-MB-231 cells were incubated with 5 mM fructose, and cell lysates were subjected to immunoprecipitation with PRPS1 antibody (or IgG), and immunoblotted with anti-KHK, or anti phospho-S/T antibody.

10. If leucine 83 of KHK-A is involved in binding to both PRPS1 and KPNB1/LRRC59, how does fructose regulate the binding of to KHK-A to PRPS1 versus to KPNB1/LRRC59?

Answer: Given the above results for the revision point #9, fructose does not affect the interaction between KHK-A and PRPS1. Yet, we do not know how fructose reinforces the

interaction between KHK-A and KPNB1/LRRC59 in the present time. To understand the differences in fructose effect on the KHK-A binding to PRPS1 and KPNB1/LRRC59, we need to carefully review the 3D molecular structures. This remains to be investigated in the future.

11. Fig. 5. KHK-C should be included as a control in the protein kinase assay.

Answer: Recombinant His-KHK-C and GST-YWHAH peptides were reacted in the kinase reaction buffer with or without fructose. As expected, YWHAH was not phosphorylated by KHK-C.

Revision: Supplementary Fig. 9d. Lines 14-15 on p9; Line 13 on p22.

12. Fig. 5C shows that fructose inhibits KHK-A-dependent phosphorylation of YWHAH.

12. The IC₅₀ and K_i of fructose on inhibition of KHK-A-mediated phosphorylation of YWHAH should be measured. What is the nuclear concentration of fructose when the cells treated with fructose? Will KHK-A be inhibited by nuclear fructose under this concentration?

Answer: As suggested, we examined the enzyme kinetics for the fructose effect on the KHK-A phosphorylation of YWHAH. The *K_m* value of KHK-A for YWHAH was 246 nM. The IC₅₀ and K_i values of fructose against the KHK-A phosphorylation of YWHAH was 0.91 mM and 0.72 mM, respectively. We also quantified the nuclear concentration of fructose. When breast cancer cells were incubated with 5 mM fructose, fructose was substantially detected in the cytosolic fraction, but not in the nuclear fraction, which indicates that fructose hardly enters into the nucleus. These results suggest that the fructose inhibition of YWHAH phosphorylation does not occur in the nucleus.

Revision: Supplementary Fig 9e, 9f, 9g, and 9h. Lines 17-25 on p9, and Line 1 on p10; Lines 12-25 on p27, Lines 1-22 on p28.

13. 14-3-3 η (YWHAH) is also a cytosolic protein. Why is KHK-A unable to phosphorylate YWHAH in cytosol?

Answer: This is a very reasonable question. In many cases, protein modifying enzymes require some scaffold partners for its specific reaction with their substrate proteins. Now we do not precisely know whether some subunits are essential for the KHK-A reaction. If so, the KHK-A reaction with YWHAH may be activated in the presence of some scaffold proteins exclusively existing in the nucleus. This is an open question to be solved.

14. Fig. 5G. Expression of Flag-tagged YWHAH proteins only limitedly restored the expression of YWHAH. Why did the cells with Flag-tagged WT YWHAH expression have much more cell invasion than their parental cells?

Answer: We understand that the western blot in Fig. 5g raised such a concern. This problem arose from the different sensitivity of antibodies against the same samples. Thus, we replaced the YWHAH blots in fig 5g and supplementary Fig 9k with others. In these results, we detected the endogenous YWHAH and ectopically expressed Flag-YWHAH in the same blots. The blots clearly showed that Flag-tagged YWHAH is much more expressed than endogenous YWHAH.

Revision: Fig 5g and Supplementary Fig 9k.

15. Fig. 6. Does D220 in PRPS1 abolish the binding of PRPS1 to KHK-A and PRPS1 phosphorylation by KHK-A?

Answer: A molecular dynamics analysis showed that a region (aa.100–114) of KHK-A closely contacts with D220 in PRPS1 (bottom in Fig. 7d). Theoretically, the mutation of D220 is expected to inhibit the interaction between PRPS1 and KHK-A. However, we think we do

not have to test this possibility in the present study. Actually, that is definitely out of our interest. This possibility could be tested by the original research team to study PRPS1.

16. Fig. 7. Does expression of YWHAH S25A blocked tumor metastasis in mouse?

Answer: For *in vivo* evaluation of breast cancer metastasis, murine MTV-TM-011-Luc-YWHAH_WT and YWHAH_S25A cells were injected orthotopically into the mammary fat pads of mice. Our results show metastatic tumor were detected in the lung, liver, spleen, kidney, brain and GI tract in YWHAH_WT expressing mice. However, in YWHAH_S25A bearing mice, metastatic tumors were hardly detected. Histological examination also showed chest metastasis rarely developed in YWHAH_S25A bearing mice, compared to YWHAH_WT bearing mice. Collectively, it is concluded that the KHK-A-phosphorylated residue, S25 of YWHAH is essential for the fructose-induced metastasis in breast cancer.

Revision: Fig. 6. Lines 15-25 on p10, and Lines 1-3 on p11; Lines 1-10 on p20, Lines 15-20 on p20; Lines 5-18 on p36.

17. A recent study (Xu et al., Science Advances, 2019) showed that KHK-A phosphorylates p62 promote tumor cell survival. Does adding fructose in medium alter p62 phosphorylation by KHK-A?

Answer: I am sorry that I could not understand such a question. We have not performed any experiments related with p62. In fact, this point is absolutely out of our interest. We have a certain hypothesis and we test it, which is the science. We cannot test everything that have been done by other research groups. I think this possibility would be better tested by the original research team that are interested in p62.

Reviewer #2

In this manuscript, the authors described a new mechanism by which KHK-A mediates the fructose-induced epithelial-mesenchymal transition and invasion of breast cancer cells. Mechanistically, KHK-A interacts with LRRC59 and KPNB1 in fructose dependent manner, and thereby is translocated into nucleus where it phosphorylates YWHAH. Phosphorylated YWHAH recruits Slug to the promoter region and represses the transcriptional activity of CDH1 gene. The experiments are well-designed with rigorous controls, and the data well support their hypothesis. However, the clinical evidence is lacking and the physiological significance of KHK-A signaling is not fully addressed, which are needed to better justify the translational potential of their findings.

Major comments:

1) The relationship between fructose consumption and metastatic progression of breast cancer has not been observed or reported, though such connection has been studied on mouse models. The lacking of clinical evidence of the fructose/KHK-A/YWHAH axis in promoting breast cancer metastasis is the major weakness of this manuscript. To justify the clinical importance of their findings, it will be better to investigate the relationship among KHK-A expression, YWHAH phosphorylation and metastasis progression in patient samples.

Answer: To examine the association of KHK-A and S25-phosphorylated YWHAH, we immunologically stained human breast cancer arrays. The cancer specimens were histologically graded according to the Nottingham grading system. The nuclear KHK-A and S25-phosphorylated YWHAH levels both were significantly higher in the grade 3 group than in the grade 2 group. Even when the breast cancer specimens were divided into non-metastasis and metastasis groups, the nuclear KHK-A and S25-phosphorylated YWHAH levels were much higher in the metastasis group. Spearman correlation analysis showed

that nuclear KHK-A expression correlates with the S25-phosphorylated YWHAH expression. Collectively, the clinical evidence supports our notion that the KHK-A-mediated phosphorylation of YWHAH is associated with breast cancer metastasis.

Revision: Fig 9a, 9b, 9c, and 9d. Lines 13-24 on p14; Lines 1-10 on p29; Lines 9-16 on p39.

2) The major findings of fructose/KHK-A induced EMT, such as down-regulation of both protein and mRNA level of CDH1, and ChIP assay of SLUG binding on CDH1 promoter, were test only on MDA-MB-231 cells. MDA-MB-231 cells are well known to be an invasive breast cancer cell line with extremely mesenchymal phenotype. It is not common to further induce EMT in such cells. In addition, EMT is known to lead to the loss of epithelial markers accompanied with gain of mesenchymal markers. However, as shown in Figure 1f, neither fructose treatment alone or with over-expressed KHK-A could increase the expression levels of mesenchymal markers in MDA-MB-231 cells. Therefore, another epithelial cell line, such as MCF7, is required to validate the effects of fructose/KHK-A on EMT.

Answer: We performed new experiments to confirm KHK-A/fructose effect on EMT in MCF-7 breast cancer cell line.

Revision: Supplementary Fig. 1d, 11a, 11c, 11e, 11h, 11i, 11k, and 11m.

3) The current design of in vivo study is less clinically relevant. The metastasis promoting effect of fructose drink was only observed in the group with ectopic expression of KHK-A but not in the control group with the physiological level of KHK-A. It would be better to include another design showing that loss of function of KHK-A could diminish the metastasis in vivo.

Answer: As suggested, we performed new experiments. First of all, I would like to mention that the fructose-induced metastasis was not only observed in KHK-A overexpressing breast tumors, but also in control ones. Given Supplementary figures 5b, 6e, and 10b, the lung

metastasis by fructose intake was also detected in the control tumors. However, when endogenous KHK-A was silenced or replaced with ectopic KHK-C, the fructose intake failed to promote tumor metastasis.

Revision: Supplementary Fig. 6 and Fig 10. Lines 12-13, 15, 19-20 and 23-24 on p6. Lines 1-6 on p7, Line 21 on p10; Lines 15-20 on p20.

Minor comments:

1) Figure 1b, the absolute quantitation qRT-PCR of KHK isoforms should be used to compare KHK-A and KHK-C expression levels in cells. The relative quantitation qRT-PCR may be misleading due to the difference in the amplification efficiency of KHK-A and KHK-C primers.

Answer and Revision: We performed qRT-PCR with newly synthesized primers targeting KHK-A or KHK-C in various cancer cell lines. Except for renal cancer cell line 786O, all cancer cell lines predominantly expressed KHK-A compared to KHK-C. The results were represented with previous qRT-PCR data in Fig 1b.

2) MDA-MB-231 cells are typically spindle shaped morphology in monolayer culture (<https://www.atcc.org/products/all/HTB-26.aspx#characteristics>), and have high expression level of vimentin but very weak or even non-detectable protein levels of both E-Cadherin and N-Cadherin as determined by western blotting (PMID: 10545506). The current immunofluorescence and immunoblotting data are not consistent among different panels or with the above mentioned impression on MDA-MB-231 cells. A cell authentication report may be required to prove the identity of MDA-MB-231 used in this study.

Answer and Revision: Western blot results are usually influence by various factors including antibody sensitivity, dilution titer, handling of membrane, and so on. We also had

hard time to detect E-cadherin and N-cadherin, but overcame this hurdle after trying many different antibodies. In addition, the mRNA result (Fig. 8b) also support that the gene for E-cadherin is expressed, not completely suppressed. In terms of the identity of MDA-MB-231, we here attached the STR profile report provided by The Korea Cell Line Bank (Seoul, South Korea). The result shows that the cell line used in this study has the same STR profile to that presented at ATCC, as follows. The ATCC STR profile for MDA-MB-231 (ATCC HTB-26) is, Amelogenin: X; CSF1PO: 12, 13; D13S317: 13; D16S539: 12; D5S818: 12; D7S820: 8, 9; TH01: 7,9,3; TPOX: 8, 9; vWA: 15, 18)

STR Profile Results

Sample	D8S1179	D21S11	D7S820	CSF1PO	D3S1358	TH01	D12S317	D16S539
NCI-H1299(control)	10,13	32,2	10	12	17	6,9,3	12	12,13
NCI-H1299	10,13	32,2	10	12	17	6,9,3	12	12,13
THP-1(control)	10,14	30,31,2	10	11,13	15,17	8,9,3	13	11,12
THP-1	10,14	30,31,2	10	11,13	15,17	8,9,3	13	11,12
MDA-MB-231(control)	13	30,33,2	8,9	12,13	16	7,9,3	13	12
MDA-MB-231	13	33,2	8,9	12,13	16	7,9,3	13	12
COLO205(control)	9,14	30,2,33,2	9,10	11,12	16	8,9	10,12	12,13
COLO205	9,14	30,2,33,2	9,10	11,12	16	8,9	10,12	12,13
786-O	13	29,30	11,12	10	16	6,9,3	8	12

Sample	D2S1338	D19S433	Vwa	TPOX	D18S51	Amelogenin	D5S818	FGA
NCI-H1299(control)	23,24	14	16,18	8	16	X	11	20
NCI-H1299	23,24	14	16,18	8	16	X	11	20
THP-1(control)	17,18	12,2,13	16	8,11	13,14	X,Y	11,12	24,25
THP-1	17,18	12,2,13	16	8,11	13,14	X,Y	11,12	24,25
MDA-MB-231(control)	20,21	11,14	15,18	8,9	11,16	X	12	22,23
MDA-MB-231	20,21	11,14,16	15,18	8,9	16	X	12	22,23
COLO205(control)	17,18	13,14	15	11	18	X	10,13	23
COLO205	17,18	13,14	15	11	18	X	10,13	21,23
786-O	17,18	14,15	15,17	8,11	13,14	X,Y	9	24

STR Kit : AmpliFLSTR identifier PCR Amplification kit (Applied Biosystems, Foster, CA, cat.4322288)

Analysis Methods : 3730 DNA Analyzer (Applied Biosystems, Foster, CA)

: GeneMapper ID v3.2 (Applied Biosystems, Foster, CA)

The fingerprinting of cell lines by 'AmpliFLSTR identifier PCR Amplification kit' was tested on September 13, 2013.

The Korean cell line bank guaranteed the authenticity of cell lines.

Korean Cell Line Bank
Cancer Research Institute
Seoul National University College of Medicine
Seoul, Korea
<http://cellbanksnu.ac.kr>
Tel : 82-2-3668-7915, Fax : 82-2-742-0021

Ja-Lok Ku, D.V.M, Ph.D.
Curator
Korean Cell Line Bank
E-mail : kujalok@snu.ac.kr

3) Several references are not properly cited. For example, neither literature 5 nor 6 discussed the association of fructose consumption and risk of multiple types of cancer except the endometrial cancer. Literature 18 does not report that breast cancer cells can metastasize in response to fructose.

Revision: Thank you for your carefully reviewing this manuscript. We replaced the references with appropriate ones in a new version manuscript.

4) Both YWHAH and KHK-A proteins are also present in the cytosolic compartment, as suggested by Figure 3D. Thus, the figure 8 is not accurate and kind of misleading. The mechanism underlying the observation that KHK-A does not interact with cytosolic YWHAH upon fructose treatment may illuminate new strategies to block this signaling axis.

Revision: We revised the graphical summary in Fig. 10, as suggested.

Reviewer #3

The paper investigates the molecular mechanisms at the basis of fructose intake and breast cancer metastasis, supporting clinical evidence for fructose-induced cancer aggravation. The authors present evidences that KHK-A signaling pathway is involved in fructose-induced cell invasion. In particular, they show molecular experiments supporting the evidence that ketohexokinase-A, rather than ketohexokinase-C, is necessary and sufficient for fructose-induced cell invasion.

While I find the paper interesting and well organised, with a nice combination of computational and experimental results, I have some major reservations that, I think, are in the way for a clear conclusion on the reported data.

1) The evidence that the main target is KHK-A and not other isoforms is based on observed overexpression of KHK-A in a number of relevant cell lines, but should be better supported.

Answer: To clarify distinct function of KHK-A from KHK-C in breast cancer metastasis, we additionally performed experiments with breast cancer cell line overexpressing KHK-C both *in vitro* and *in vivo*:

Revision: Fig 1d, 1e, Supplement Fig 3f, 3g, 6, 7b, 7c, 8b, 8e, and 9d. Lines 5-8 on p5 (related to Fig 1d, 1e); Lines 20-24 on p5 (related to Supplementary fig 3f, 3g); Lines 12-13, 15, 19-20, 23-24 on p6, Lines 1-6 on p7 (related to supplementary fig 6); Lines 24-25 on p7 (related to supplementary fig 7b, c); Lines 10-11 on p8 (related to supplementary fig 8b, e); Lines 11-12 on p9 (related to supplementary fig 9d)

2) A recent paper (not cited by the authors) Clin Invest. 2018;128(6):2226-2238. present convincing evidence that protection from High Fructose Intolerance (HFI) is resulting from inhibition of the KHK-C isoform, and not the A. So the mechanistic interplay of these two Kinases seems to have a prominent role with respect to fructose intake. Once silenced KHK-A what happens to the expression on KHK-C *in vivo*? We do not know if these isoforms would replace the KHK-A activity in such condition and therefore we cannot use the argument of specifically inhibiting KHK-A. Moreover how specific is the named inhibitor is not clarified and not texted here with respect to the fructose signalling pathways. The authors should show the results also with less specific inhibitors, and if available, inhibitors targeting KHK-C vs KHK-A.

Answer: To answer the first question, we tested whether the expression level of KHK-A can affect KHK-C expression level under the presence of fructose in two breast cancer cell lines. The results show that the expression of KHK-C does not increase even after KHK-A knock-down or fructose treatment (below panel). KHK-C may not compensate the downregulation

of KHK-A.

MDA-MB-231 and MCF-7 cells which had been silenced with si-*KHK-A*, were incubated with 5 mM fructose for indicated time point. Total lysates were subjected to western blotting. HepG2 cell line was used as positive control for KHK-A and KHK-C.

For the second question, the inhibitor used in this project is pyrimidinopyrimidine (also known as KHK inhibitor, sigma, 420640). According to the information sheet, this comical is a reversible and ATP-competitive inhibitor. IC50 against KHK is so low as 12 nM, which may support its specificity to KHK. The inhibitor targets the ATP binding motif, which is 100% conserved in KHK-A and KHK-C, so it inhibits both isoforms with the same efficiency. As far as I know, unfortunately, the isoform-specific inhibitors have not been available yet.

3) Major revision

I have some major reservations also on the computational data here reported.

The docking and pulling simulations are very poorly described. The intent of the pulling simulation was to describe the Potential of mean force (PMF) associated to the process of S25-OH approaching ATP-gammaPO3 and therefore support the hypothesis that YWHAH is phosphorylating KHK-A (Fig 6 panel a). First of all the description of the residue contacts playing as role in this process is difficult to follow and the comparison with PRPS11 is not clearly discussed and documented. Moreover the PMF is reported as a single profile for each experiment and there is no evidence of the reproducibility or of the error associated to these. Multiple replica of the pulling process are usually needed to validate the observations.

Additionally, resampling with the WHAM method, for example, would at least give some confidence to the observed overlap or differences in the profiles.

Answer: Upon the reviewer's comments, we added more descriptions for docking and pulling simulations (in Method section for Molecular dynamics simulation) and elaborated more on the discussion for the comparison with PRPS11 and for the clarification of contact map. Also, following the reviewer's comments on the reproducibility of PMF, we performed 5 fully independent sets of umbrella simulations for each of profiles to check the reproducibility. The uncertainty of each profile was also estimated by resampling technique using bootstrap analysis with 200 bootstraps.

Revision: Fig. 7a. Lines 2-11 on p12.

REVIEWER COMMENTS

Reviewer #1 (Remarks to the Author):

The authors have successfully addressed most of specific points, but a few questions (such as Points 13, 17) about the connection of current findings with previously published functions of KHK-A in cancer progression are not addressed. In addition, the authors did not provide any answers to the major concerns, as list below:

“However, how fructose promotes nuclear accumulation of KHK-A but not KHK-C is not investigated. Given that KHK-A has limited binding affinity to fructose, how fructose triggers a signaling to affect KHK-A cellular redistribution is of significance to understand the role of fructose metabolism in cancer progression.”

As mentioned in the earlier comments, the authors' finding is interesting. However, the remaining questions would be obvious to the people working in the metabolism filed. The answers to these questions would largely improve the quality of the current work.

Reviewer #2 (Remarks to the Author):

The authors have sufficiently addressed my previous concerns. The manuscript is substantially improved.

Reviewer #3 (Remarks to the Author):

The authors have revised the manuscript and responded to mine and other referees queries. I am happy with the response to my first query about separating the effect of KHK-a from KHK-C in a number of breast cancer cell lines. The authors show the effect of silencing endogenous KHK-A and replacing with ectopic KHK-C on fructose intake and tumor growth and body weight convincingly.

On the other hand I think the pulling simulations and the interpretation of the data are still missing information and clarification.

The pulling coordinate is spanning 0.3/0.4 nm: this means the authors have calculated 3/4 histograms only. Can the author show these and their overlap? This information is important in judging convergence of the sampling. I am surprised that one can capture a complex profile with few histograms only (although the two pulling groups are small and quite mobile, so might cross the energy barrier more easily). So. clarification in needed here

Have the authors computed the profile using WHAM? this should be mentioned and which bootstrap method have they used?

Bayesian bootstrapping or simple bootstrapping? The latter removes one histogram at random, and it would be inappropriate with so few histograms.

The plots are reported in nm for the distance, but the figure legend and some test is still referring to distances in Angstrom, consistency should be adopted.

Also the comment I had on another paper and the isoform KHK-C playing a role has been dismissed by the Authors, I think with the new evidence reported here and a better discussion of the relative role of KHK-A vs KHK-C in this context should be given and elaborated in the context of the Clin Invest. 2018;128(6):2226-2238 paper.

I find in general the paper misses a strong discussion/conclusion that organically merges all the different evidences and their importance in the characterisation of the specific role of KHK-A in mediating fructose-induced metastasis in breast cancer.

There are a number of typos to be corrected (photones instead of photons as an example) and the form should be revised as it is sloppy at points.

REVIEWERS' COMMENTS AND ANSWERS

Reviewer #1

The authors have successfully addressed most of specific points, but a few questions (such as Points 13, 17) about the connection of current findings with previously published functions of KHK-A in cancer progression are not addressed. In addition, the authors did not provide any answers to the major concerns, as list below: “However, how fructose promotes nuclear accumulation of KHK-A but not KHK-C is not investigated. Given that KHK-A has limited binding affinity to fructose, how fructose triggers a signaling to affect KHK-A cellular redistribution is of significance to understand the role of fructose metabolism in cancer progression.” As mentioned in the earlier comments, the authors’ finding is interesting. However, the remaining questions would be obvious to the people working in the metabolism filed. The answers to these questions would largely improve the quality of the current work.

Q1, 14-3-3 η (YWHAH) is also a cytosolic protein. Why is KHK-A unable to phosphorylate YWHAH in cytosol?

Answer: Honestly speaking, we do not know the reason yet. We could speculate several possibilities. As shown in Supplementary Figure 9, the KHK-A phosphorylation of YWHAH is inhibited by fructose because fructose may compete with YWHAH for KHK-A binding. If this event occurs within cells, such an action of KHK-A could be inhibited in the cytoplasm where fructose is enriched. On the contrary, KHK-A may get the activity in the nucleus where fructose exists at a lesser level. As other possibilities, we could speculate that the KHK-A-YWHAH interaction is interfered by some cytoplasmic proteins or enforced by some nuclear proteins. Alternatively, KHK-A and/or YWHAH are differently modified in the cytoplasm and nucleus, which may affect the specific activity of KHK-A to YWHAH. Anyway, we now do not know the precise mechanism and this is an open question. I hope it will be understood that

this work is limited in some points.

Q2. A recent study (Xu et al., Science Advances, 2019) showed that KHK-A phosphorylates p62 promote tumor cell survival. Does adding fructose in medium alter p62 phosphorylation by KHK-A?

Answer: As suggested, we tested the S28-phosphorylation of p62 under fructose in Hep3B, MCF-7 and MDA-MB-231. The results showed that p62-S28 is phosphorylated under hypoxic conditions in these cells, but not under 5 mM fructose.

Revision: Supplementary Fig 10 and its legend; the last sentence on p10 and the first sentence lines on p11.

Q3. “However, how fructose promotes nuclear accumulation of KHK-A but not KHK-C is not investigated. Given that KHK-A has limited binding affinity to fructose, how fructose triggers a signaling to affect KHK-A cellular redistribution is of significance to understand the role of fructose metabolism in cancer progression.”

Answer: First of all, I would like to note the meaning of substrate affinity in the enzymology. We can know the affinity from the Km value (Affinity = $1/K_m$). In fact, the affinity used in the enzyme kinetics is quite different from the physically binding affinity (Affinity = $1/K_d$) between the enzyme and substrate. To answer this question, we used radioactive fructose and measured the physical affinities of fructose to KHK-A and KHK-C. In HEPES and phosphate buffers, the Kd values of KHK-A for fructose are 0.53 and 0.57 mM, respectively. In the same buffers, the Kd of KHK-C for fructose are 0.40 and 0.45 mM, respectively (Supplementary Fig, 8f). Surprisingly, KHK-A and KHK-C have the similar affinity values ($1/K_d$) in fructose binding. It is so true that KHK-C has a higher (enzymatic) activity in fructose phosphorylation than KHK-A. But we think that KHK-A is sensitive to fructose as

much as KHK-C. Perhaps, the K_m difference in the enzyme kinetics could result from the molecular distance between fructose and ATP within the catalytic pockets of KHK-A and C. Indeed, the fructose binding domain is just same in KHK-A and KHK-C because the exon 3 segment (isotope-specific part) does not participate in fructose binding. This information also supports our notion.

Revision: Supplementary figure 8f; the 2nd paragraph on p8, the 1st paragraph on p30.

Reviewer #3

The authors have revised the manuscript and responded to mine and other referees queries. I am happy with the response to my first query about separating the effect of KHK-a from KHK-C in a number of breast cancer cell lines. The authors show the effect of silencing endogenous KHK-A and replacing with ectopic KHK-C on fructose intake and tumor growth and body weight convincingly.

Q1. On the other hand, I think the pulling simulations and the interpretation of the data are still missing information and clarification. The pulling coordinate is spanning 0.3/0.4 nm: this means the authors have calculated 3/4 histograms only. Can the author show these and their overlap? This information is important in judging convergence of the sampling. I am surprised that one can capture a complex profile with few histograms only (although the two pulling groups are small and quite mobile, so might cross the energy barrier more easily). So, clarification is needed here. Have the authors computed the profile using WHAM? this should be mentioned and which bootstrap method have they used? Bayesian bootstrapping or simple bootstrapping? The latter removes one histogram at random, and it would be inappropriate with so few histograms. The plots are reported in nm for the distance, but the figure legend and some text is still referring to distances in Angstrom, consistency should be

adopted.

Answer: Thank you so much for your critical question. Upon the reviewer's comments, we added histograms obtained for computing PMF profiles. As can be seen in the figures below, the histograms from the umbrella sampling show reasonably sufficient overlap between neighboring windows despite only 4 windows used for the sampling. As pointed out by Reviewer, this seems to be due to the fact that the pulling groups are quite mobile such that they can cross the energy barrier easily, as also indicated from the bimodal distributions found in some histogram windows.

Regarding Reviewer's comments on the method for computing the PMF profiles and bootstrap method, we used WHAM with Bayesian bootstrapping, which was mentioned in Materials and Method section of the revised manuscript. Also, as for the length unit of the distance (d) between two pulling groups, we revised all the length units in the figure captions and legends in nm for consistency.

Revision: the 1st paragraph on p32.

Q2. Also the comment I had on another paper and the isoform KHK-C playing a role has been dismissed by the Authors, I think with the new evidence reported here and a better discussion of the relative role of KHK-A vs KHK-C in this context should be given and elaborated in the context of the Clin Invest. 2018;128(6):2226-2238 paper. I find in general the paper misses a strong discussion/conclusion that organically merges all the different evidences and their importance in the characterisation of the specific role of KHK-A in mediating fructose-induced metastasis in breast cancer.

Answer: As recommended, we further discussed the roles of KHK-A and KHK-C. "The KHK-C-driven fructose metabolism has been intensively investigated related with the protein kinase activity of KHK-A."

Revision: the 2nd paragraph on p16

Q3. There are a number of typos to be corrected (photones instead of photons as an example) and the form should be revised as it is sloppy at points.

Answer: Thank you so much for your carefully reviewing this manuscript. We corrected text errors in the manuscript.

REVIEWERS' COMMENTS:

Reviewer #1 (Remarks to the Author):

The remaining concerns have been addressed

Reviewer #3 (Remarks to the Author):

The Authors have addressed my queries satisfactorily.